# Coordination of bacterial cell wall and outer membrane biosynthesis

Katherine R. Hummels[1], Samuel P. Berry[2], Zhaoqi Li[1], Atsushi Taguchi[1,3], Joseph K. Min[2], Suzanne Walker[1], Debora S. Marks[2] & Thomas G. Bernhardt[1,4 ✉]

Gram-negative bacteria surround their cytoplasmic membrane with a peptidoglycan (PG) cell wall and an outer membrane (OM) with an outer leaflet composed of lipopolysaccharide (LPS)[1]. This complex envelope presents a formidable barrier to drug entry and is a major determinant of the intrinsic antibiotic resistance of these organisms[2]. The biogenesis pathways that build the surface are also targets of many of our most effective antibacterial therapies[3]. Understanding the molecular mechanisms underlying the assembly of the Gram-negative envelope therefore promises to aid the development of new treatments effective against the growing problem of drug-resistant infections. Although the individual pathways for PG and OM synthesis and assembly are well characterized, almost nothing is known about how the biogenesis of these essential surface layers is coordinated. Here we report the discovery of a regulatory interaction between the committed enzymes for the PG and LPS synthesis pathways in the Gram-negative pathogen *Pseudomonas aeruginosa*. We show that the PG synthesis enzyme MurA interacts directly and specifically with the LPS synthesis enzyme LpxC. Moreover, MurA was shown to stimulate LpxC activity in cells and in a purified system. Our results support a model in which the assembly of the PG and OM layers in many proteobacterial species is coordinated by linking the activities of the committed enzymes in their respective synthesis pathways.

The biosynthetic pathways for phospholipids, PG and LPS in Gram-negative bacteria rely on shared precursor pools (Fig. 1a). Therefore, flux through each pathway must be balanced to prevent overconsumption of essential precursors by a single pathway[4–6]. LPS biosynthesis requires both UDP-*N*-acetylglucosamine (UDP-GlcNAc) and acyl-ACP molecules that are also used for PG and phospholipid biosynthesis, respectively[6,7]. Additionally, overproduction of LPS results in the toxic accumulation of LPS intermediates in the inner membrane[8]. Flux through the LPS pathway must therefore be tightly regulated.

Enterobacteria such as *Escherichia coli* control LPS synthesis through regulated proteolysis of the committed enzyme, *Ec*LpxC, by FtsH[4,9]. Previous studies of LpxC in *P. aeruginosa* (*Pa*LpxC), however, showed that it was not proteolysed[10]. Accordingly, N-terminally His-tagged *Pa*LpxC (H–*Pa*LpxC) accumulated normally in an *ftsH*-deletion mutant (Extended Data Fig. 1a). Thus, LPS biogenesis in *P. aeruginosa* seems to be regulated through a mechanism distinct from that in enterobacteria.

## *Pa*LpxC is activated by *Pa*MurA

The overproduction of H–*Ec*LpxC but not H–*Pa*LpxC inhibited growth of *P. aeruginosa* and increased cellular levels of LPS (Extended Data Figs. 1b and 2a), suggesting that *Pa*LpxC is regulated in *P. aeruginosa* cells through a mechanism ineffective against *Ec*LpxC. To identify possible regulatory factors, H–*Pa*LpxC or H–*Ec*LpxC was produced in *P.*

*aeruginosa* and interaction partners were identified following affinity purification. *Pa*MurA and PA4701 were the only proteins enriched with H–*Pa*LpxC but not H–*Ec*LpxC (Supplementary Table 1). PA4701 is a non-essential protein (Extended Data Fig. 3a) of unknown function, whereas *Pa*MurA is the essential committed enzyme for PG synthesis[11,12] (Fig. 1a and Extended Data Fig. 3b,c). We focused on the *Pa*MurA–*Pa*LpxC interaction and validated it using in vivo pulldown assays with H–*Pa*LpxC and N-terminally Flag-tagged *Pa*MurA (F–*Pa*MurA; Extended Data Fig. 4a). Notably, the co-purification was enhanced when cells were treated with the LpxC inhibitor CHIR-090, suggesting that the drug-bound LpxC enzyme may have a greater affinity for MurA (Extended Data Fig. 4a). Purified His-tagged *Pa*MurA (H–*Pa*MurA; Extended Data Fig. 5a) was also specifically pulled down by Flag-tagged *Pa*LpxC (F–*Pa*LpxC) using anti-Flag resin, indicating that the interaction was direct (Extended Data Fig. 4b). Purified F–*Pa*LpxC and H–*Pa*MurA were subjected to size-exclusion chromatography (SEC) individually or as 1:1 mixtures with or without CHIR-090. In the mixed sample, a large proportion of protein eluted at a volume corresponding to the heterodimer of F–*Pa*LpxC and H–*Pa*MurA (Fig. 1b). Consistent with the pulldown assays, re-application of the heterodimer fractions to the SEC column showed that the complex remained most stable in the presence of CHIR-090 (Fig. 1b). Notably, H–*Pa*MurA stimulated the enzymatic activity of F–*Pa*LpxC (Fig. 1c). This stimulation was specific to *Pa*MurA as the addition of H–*Ec*MurA had no effect (Fig. 1c). We conclude that *Pa*MurA is a direct and specific activator of *Pa*LpxC in vitro.

[1]Department of Microbiology, Blavatnik Institute, Harvard Medical School, Boston, MA, USA. [2]Department of Systems Biology, Blavatnik Institute, Harvard Medical School, Boston, MA, USA. [3]SANKEN (The Institute of Scientific and Industrial Research), Osaka University, Ibaraki, Japan. [4]Howard Hughes Medical Institute, Boston, MA, USA. ✉e-mail: thomas_bernhardt@hms.harvard.edu

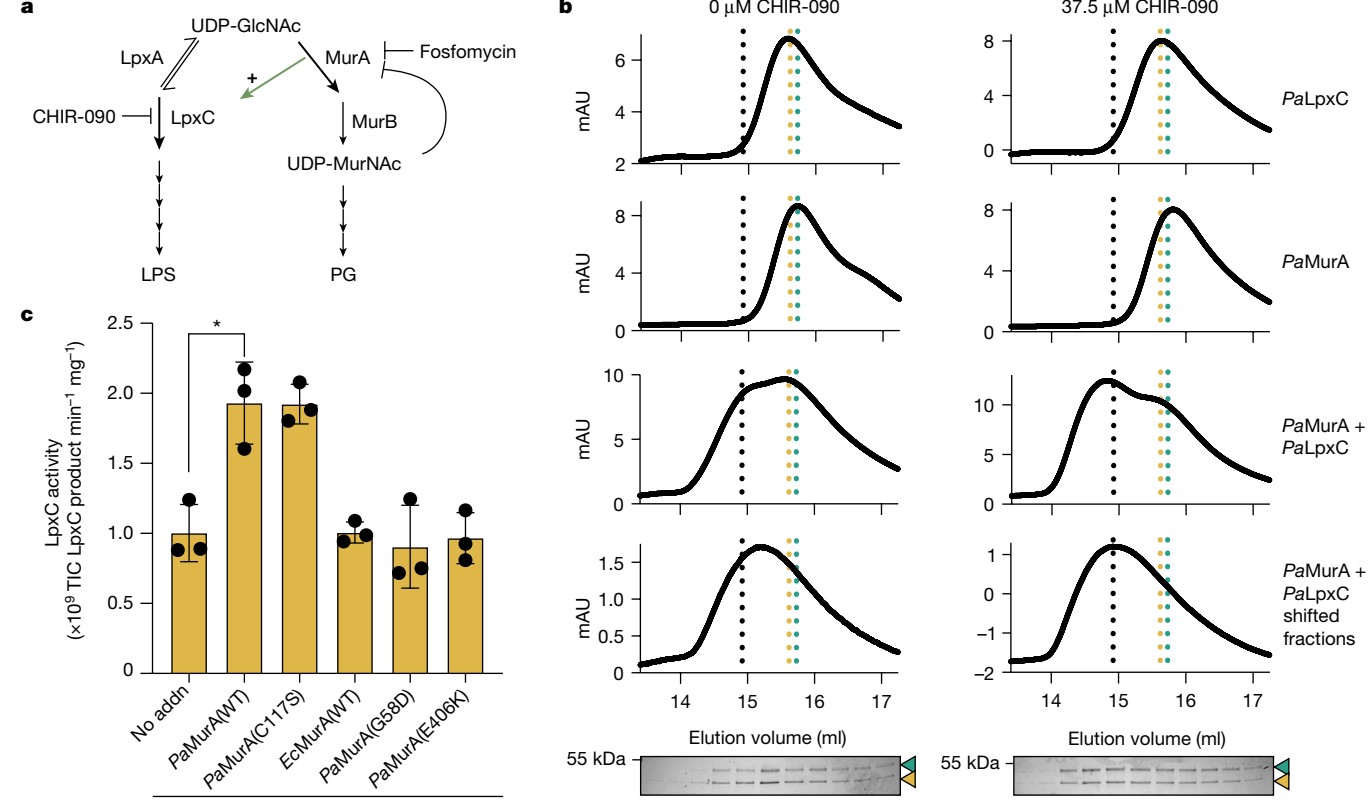

**Fig. 1 | *Pa*MurA interacts with and activates *Pa*LpxC. a**, Schematic representation of the biosynthetic pathways responsible for PG and LPS biosynthesis, showing the committed enzymes MurA and LpxC, and other relevant enzymes (LpxA and MurB). T-bars indicate inhibition by the antibiotics fosfomycin and CHIR-090. The green arrow indicates activation of LpxC by MurA. **b**, Size-exclusion chromatography in which 7.5 µM of purified F–LpxC and H–MurA were resolved either individually or as a mixture in the presence or absence of 37.5 µM CHIR-090 as indicated. The shifted F–LpxC + H–MurA fractions were subsequently collected, resubjected to size-exclusion chromatography, and the resulting fractions were resolved by SDS–PAGE followed by Coomassie staining. Dotted lines indicate the peak mobilities for F–LpxC (gold), H–MurA (cyan) or the shifted F–LpxC + H–MurA fractions (black) in the presence of CHIR-090. The mobilities of H–MurA and F–LpxC in SDS–PAGE assays are indicated by a cyan or gold arrowhead, respectively. Data are representative of two replicates. For gel source data, see Supplementary Fig. 1. **c**, Catalytic activity (total ion counts (TIC)) of purified *Pa*LpxC (100 nM) alone (no addition (No addn)) or in the presence of MurA variants (100 nM). Dots indicate the values obtained for three individual replicates, bars indicate the mean, and error bars represent their standard deviation. \*$P = 0.0109$ (unpaired, two-tailed *t*-test).

We reasoned that if *Pa*MurA is an essential activator of *Pa*LpxC in *P. aeruginosa* cells, then *Pa*MurA depletion should result in a phenotype that resembles the simultaneous inactivation of both essential enzymes. Cells treated with the MurA inhibitor fosfomycin have a PG synthesis inhibition phenotype involving membrane bleb formation and lysis[11] (Extended Data Fig. 3d–f). However, cells depleted of *Pa*MurA instead adopted an enlarged, ovoid shape (Extended Data Fig. 3e,f). This phenotype resembled that of cells treated with both CHIR-090 and fosfomycin (Extended Data Fig. 3d–f), suggesting that *Pa*MurA depletion impairs the synthesis of both PG and LPS. Accordingly, *Pa*MurA depletion reduced LPS levels, whereas depletion of the next enzyme in the pathway, *Pa*MurB, did not (Extended Data Figs. 2b and 3b,c) and instead caused a terminal phenotype resembling that following fosfomycin treatment (Extended Data Fig. 3d–f). We conclude that *Pa*MurA is required for the biosynthesis of normal cellular levels of LPS.

## *Pa*MurA has two essential functions

Overproduction of wild-type (WT) *Pa*MurA did not increase LPS levels as expected for an activator of *Pa*LpxC (Extended Data Fig. 2c). We suspected this result might be due to the enzymatic activity of *Pa*MurA competing with the LPS pathway for UDP-GlcNAc (Fig. 1a) thereby preventing runaway LPS synthesis. Alternatively, a particular MurA conformer[13] may be the activator and it may not be produced in sufficient

levels to stimulate LPS synthesis on *Pa*MurA overexpression. These scenarios predict that the overproduction of catalytically inactive or conformationally trapped *Pa*MurA variants should cause lethal levels of LPS production. We therefore searched for toxic *Pa*MurA variants. *P. aeruginosa* was transformed with a plasmid encoding mutagenized P*murA* under the control of an isopropyl-β-ᴅ-thiogalactoside (IPTG)-regulated promoter. The resulting library was pooled and grown in liquid medium with inducer. Plasmids that caused lysis were purified from the culture supernatant, and the P*murA* genes from those causing an IPTG-dependent growth defect on retransformation were sequenced.

Twenty-three toxic *Pa*MurA variants (designated *Pa*MurA\*) were identified (Fig. 2a and Extended Data Fig. 6a). Their growth inhibitory activity was alleviated by a normally lethal concentration of CHIR-090, suggesting that they hyperactivate *Pa*LpxC (Extended Data Fig. 6a). The amino acid changes in the *Pa*MurA\* variants mapped around the active site of the enzyme[14] (Extended Data Fig. 6b) and included the catalytic cysteine residue (C117) that forms a covalent intermediate with the phosphoenolpyruvate substrate[15]. A subset of *Pa*MurA\* variants were purified (Extended Data Fig. 5a) and found to have markedly reduced enzymatic activity while retaining their ability to activate purified *Pa*LpxC (Extended Data Fig. 6c,d).

The equivalent of the C117S substitution in *Pa*MurA has been well characterized in other orthologues in which it has been shown to trap the enzyme in a closed conformation bound to its product[16,17].

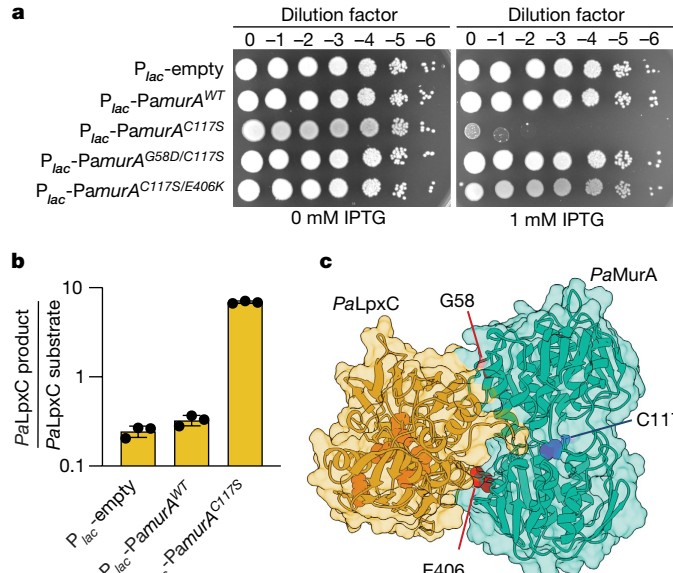

**Fig. 2 | *Pa*MurA(C117S) is a potent activator of *Pa*LpxC in vivo. a**, Viability assay in which serial dilutions of PAO1 harbouring an empty plasmid or the indicated P*amurA* variant under IPTG-inducible control were plated on LB agar with or without IPTG supplementation as indicated. Data are representative of three biological replicates. **b**, Ratio of LpxC product to substrate detected by liquid chromatography–mass spectrometry in the aqueous fraction of methanol–chloroform-extracted *P. aeruginosa* whole-cell lysates. Dots indicate the values obtained for three biological replicates, bars indicate the mean, and error bars represent their standard deviation. **c**, Model structure of the *Pa*LpxC–*Pa*MurA complex predicted by AlphaFold[19]. *Pa*LpxC is represented in gold and *Pa*MurA is represented in cyan. *Pa*MurA residues G58 and E406 are highlighted in red and residue C117 is highlighted in navy. *Pa*LpxC active-site residues[34] are depicted in orange.

We therefore chose to study the potential in vivo activation of *Pa*LpxC by *Pa*MurA(C117S) further. The LpxC substrate (UDP-3-*O*-(*R*-3-hydroxydecanoyl)-*N*-acetylglucosamine) and product (UDP-3-*O*-(*R*-3-hydroxydecanoyl)-glucosamine) were extracted and quantified from cells overproducing *Pa*MurA(WT) or *Pa*MurA(C117S). The *Pa*LpxC product/substrate ratio was unchanged in cells overproducing *Pa*MurA(WT) relative to an empty vector control (Fig. 2b and Extended Data Fig. 7). However, overproduction of *Pa*MurA(C117S) led to a 100-fold increase in the *Pa*LpxC product/substrate ratio and increased LPS levels, indicating that this variant is a potent activator of *Pa*LpxC in vivo (Fig. 2b and Extended Data Figs. 2c and 7). We therefore conclude that *Pa*MurA activates *Pa*LpxC in cells but that only catalytically inactive *Pa*MurA variants promote toxic levels of LPS synthesis when they are overproduced owing to their inability to compete with the LPS synthesis pathway for UDP-GlcNAc.

To further characterize the interaction between *Pa*LpxC and *Pa*MurA, we searched for non-toxic derivatives of *Pa*MurA(C117S), reasoning that some would be unable to bind *Pa*LpxC. Survivors of F–*Pa*MurA(C117S) production from a mutagenized plasmid were selected followed by an immunoblot-based screen for isolates producing stable, full-length F–*Pa*MurA(C117S). This procedure identified *Pa*MurA(C117S/G58D) and *Pa*MurA(C117S/E406K). We confirmed that both variants were not toxic when overproduced to levels similar to *Pa*MurA(C117S) (Fig. 2a and Extended Data Fig. 8a). We next purified H–*Pa*MurA(G58D) and H–*Pa*MurA(E406K) variants with intact active sites (Extended Data Fig. 5a). Both retained MurA activity (Extended Data Fig. 8b), and consistent with the genetic results, they failed to activate F–*Pa*LpxC in vitro (Fig. 1c). Notably, however, whereas H–*Pa*MurA(G58D) was unable to bind to F–*Pa*LpxC, H–*Pa*MurA(E406K) exhibited binding activity

similar to that of H–*Pa*MurA(WT) (Extended Data Fig. 8c). Thus, the G58D change impairs the ability of *Pa*MurA to bind *Pa*LpxC, whereas the E406K substitution seems to disrupt the activation mechanism following complex formation. The residues G58 and E406 both lie on the same surface of the *Pa*MurA structure, suggesting that this face comprises the *Pa*LpxC-binding interface (Fig. 2c).

The results thus far suggest that *Pa*MurA may have two essential functions: PG synthesis and activation of *Pa*LpxC. This model predicts that a catalytically active variant of *Pa*MurA that cannot activate *Pa*LpxC should fail to complement a P*amurA* deletion but remain capable of complementing a catalytically dead *Pa*MurA variant that retains its *Pa*LpxC activation function. To test this prediction, P*amurA*[WT] was placed under P*lac* control and the native allele was either deleted or converted to P*amurA*[C117S]. We then introduced a plasmid with or without the P*amurA*[G58D] gene controlled by an arabinose-inducible promoter (P*ara*). As expected, *Pa*MurA(WT) depletion resulted in a severe growth defect in either the Δ*murA* or P*amurA*[C117S] backgrounds (Extended Data Fig. 8d). Notably, however, the production of *Pa*MurA(G58D) was able to restore growth on *Pa*MurA(WT) depletion in the context of the P*amurA*[C117S] allele, but not the *murA* deletion (Extended Data Fig. 8d). We therefore conclude that the *Pa*LpxC activation function of *Pa*MurA is essential and separable from its catalytic activity.

## LpxC–MurA interaction is conserved

To investigate the conservation of the LpxC–MurA regulatory interaction, we used evolutionary covariation analysis[18]. However, because both enzymes are conserved throughout Gram-negative bacteria but only a subset are likely to interact, we could not reliably detect residues that covary between LpxC and MurA without first knowing which regions of the two proteins interact; non-interacting pairs among the genomes analysed generated too much 'noise' to detect a clear interaction signature. We therefore modelled the structure of the LpxC–MurA complex with AlphaFold[19], which predicted a high-confidence structure for *Pa*LpxC–*Pa*MurA but not between the corresponding *E. coli* proteins (Fig. 2c and Extended Data Fig. 9a,b). The G58 and E406 residues of *Pa*MurA implicated in *Pa*LpxC binding and/or activation lie at the modelled interaction interface (Fig. 2c), supporting the accuracy of the structure prediction. Using this model as a guide, evolutionary covariation analysis of residues predicted to be at the interaction interface was carried out on LpxC–MurA pairs from 8,302 proteobacterial genomes. Each LpxC–MurA pair was then assigned an 'interaction score' as an indication of how strongly the two proteins were predicted to interact (Fig. 3a).

The *P. aeruginosa* LpxC–MurA pair had a high interaction score as expected, as did those from other Pseudomonadales and a subset of other gammaproteobacteria including the Legionalles, Xanthomonadales and Oceanospirillales orders (Fig. 3a). In addition, high interaction scores were also observed in a subset of Rhodospirillales, an order of alphaproteobacteria that is highly divergent from *P. aeruginosa* (Fig. 3a). On the basis of the distribution of interaction scores, we estimate that 48% of gammaproteobacteria and 35% of alphaproteobacteria encode LpxC–MurA pairs that are capable of interacting (Extended Data Fig. 9e). To investigate the accuracy of the covariation analysis, three additional LpxC–MurA pairs with high interaction scores (*Magnetospirillum kuznetsovii*, *Xanthomonadales* spp. and *Legionella pneumophila*) and two LpxC–MurA pairs with low interaction scores (*E. coli* and *Acinetobacter baumannii*) were purified (Extended Data Fig. 5a) and their ability to interact was tested in vitro. Only those pairs with high interaction scores were observed to interact (Fig. 3b). Furthermore, *L. pneumophila* LpxC (*Lpn*LpxC), but not *Ec*LpxC, exhibited increased activity in vitro in the presence of its cognate MurA (Extended Data Fig. 9f,g). Thus, we conclude that LpxC and MurA from a variety of proteobacteria interact, suggesting that the regulatory link between the PG and LPS biogenesis pathways is conserved in a variety of Gram-negative bacteria.

**a**

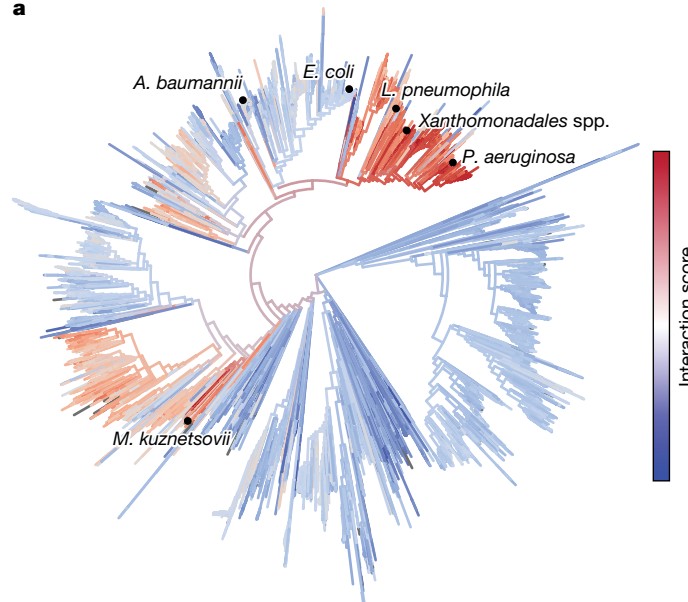

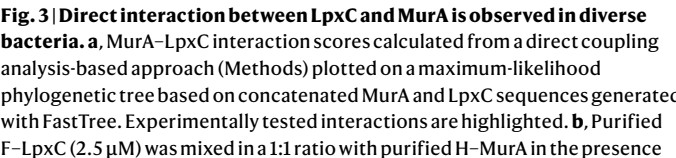

**b**

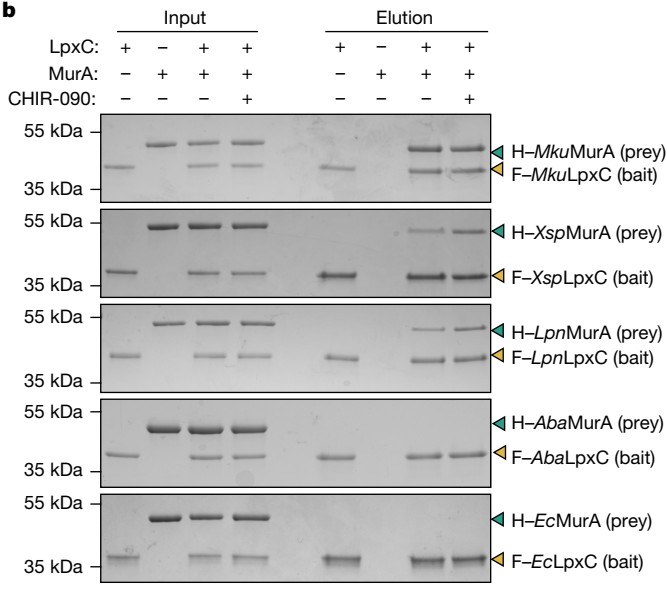

**Fig. 3 | Direct interaction between LpxC and MurA is observed in diverse bacteria. a**, MurA–LpxC interaction scores calculated from a direct coupling analysis-based approach (Methods) plotted on a maximum-likelihood phylogenetic tree based on concatenated MurA and LpxC sequences generated with FastTree. Experimentally tested interactions are highlighted. **b**, Purified F–LpxC (2.5 µM) was mixed in a 1:1 ratio with purified H–MurA in the presence or absence of CHIR-090 (5.7 µM) as indicated. The mixtures were pulled down with anti-Flag resin and the input and eluate were subjected to SDS–PAGE and Coomassie staining. Mobilities of H–MurA and F–LpxC are indicated by a cyan or gold arrowhead, respectively. Data are representative of at least two replicates. For gel source data, see Supplementary Fig. 1.

## Model for balanced LPS and PG synthesis

The LPS biosynthetic pathway shares critical precursors with both the phospholipid and PG biosynthetic pathways (Fig. 1a). In *E. coli* and other enterobacteria, the need for balanced consumption of acyl-ACP molecules by the LPS and phospholipid biosynthetic pathways has been well documented[4,20–23]. By contrast, little is known about how UDP-GlcNAc utilization is coordinated by the LPS and PG biosynthetic pathways to allow uniform cell envelope expansion. We propose that the *Pa*LpxC-activating function of *Pa*MurA serves to limit LPS biosynthesis such that it cannot outrun PG biosynthesis. In support of this model, overexpression of *Pa*MurA(WT), which is capable of competing with the LPS synthesis pathway for UDP-GlcNAc, had no impact on cellular viability (Fig. 2a). Overexpression of the catalytically inactive variant *Pa*MurA(C117S), however, resulted in the imbalanced activation of *Pa*LpxC in vivo, resulting in cell death (Fig. 2a). By limiting the catalytically active population of *Pa*LpxC to *Pa*MurA levels, runaway LPS biosynthesis is not possible. On the other hand, MurA has been shown to be feedback inhibited by the downstream PG precursor UDP-MurNAc[24]. Under conditions in which the flux of PG precursor synthesis is too high, UDP-MurNAc will accumulate such that it binds to MurA and locks it into a closed conformation similar to that of *Pa*MurA(C117S)[13,16]. Thus, when PG biosynthesis outpaces LPS biosynthesis, MurA will be subject to feedback inhibition, decreasing its catalytic activity without affecting its ability to activate *Pa*LpxC to stimulate LPS biosynthesis and rebalance the pathways.

In enterobacteria, LpxC is proteolysed by FtsH, which is in turn regulated by at least two accessory proteins: LapB and YejM[5,8,25,26]. Although FtsH is broadly conserved, LapB and YejM are observed only in a subset of proteobacterial genomes. By contrast, LpxC and MurA are nearly universally conserved in proteobacteria. Our computational analysis suggests that a substantial group of gammaproteobacteria and alphaproteobacteria control LpxC through activation by MurA (Fig. 3a). Why bacteria use different strategies to regulate flux through the LPS biosynthetic pathway is unclear. We suggest, however, that it may reflect differences in their environmental niche. For example, *P. aeruginosa* preferentially utilizes tricarboxylic acids over carbohydrates as a carbon source[27]. Thus, activated sugars such as UDP-GlcNAc may be more limiting for *P. aeruginosa* than for sugar-loving *E. coli* such that it focuses its regulation of LpxC on the appropriate partitioning of UDP-GlcNAc. It would not be surprising if future studies uncover other as-yet-unknown strategies of LpxC regulation in other organisms.

## Conclusions

In conclusion, our work has identified a previously unknown and unexpected regulatory interaction between essential enzymes involved in the production of two different layers of the Gram-negative cell envelope. In addition to uncovering a potential mechanism for coordinating the biogenesis of the cell wall and OM, these findings also have implications for antibiotic development. MurA is the target of the antibiotic fosfomycin and LpxC has been the subject of several drug discovery campaigns that have yielded inhibitors with potent antibacterial activity[11,28–33]. The connection between these enzymes identified here suggests that it may be possible to develop dual-targeting drugs that alter both MurA and LpxC activity to simultaneously disrupt PG and OM assembly to kill *P. aeruginosa* and/or sensitize it to other antibiotics made ineffective by the barrier function of its envelope. These findings also raise the possibility that there are many more regulatory interactions between enzymes involved in the biogenesis of different cell envelope components waiting to be discovered.

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

# Methods

## Media, bacterial strains and plasmids

As indicated, cells were grown in LB (1% tryptone, 0.5% yeast extract, 0.5% NaCl) or minimal M9 medium[35] supplemented with 0.2% casamino acids and 0.2% glucose. The following concentrations of antibiotics were used: chloramphenicol, 25 µg ml⁻¹; kanamycin, 25 µg ml⁻¹; gentamycin, 30 µg ml⁻¹ (*P. aeruginosa*) or 15 µg ml⁻¹ (*E. coli*); carbenicillin, 200 µg ml⁻¹ (*P. aeruginosa*) or 50 µg ml⁻¹ (*E. coli*). Plasmids used in this study are listed in Supplementary Table 2. The bacterial strains used in this study are listed in Supplementary Table 3. All *P. aeruginosa* strains used in the reported experiments are derivatives of PAO1. Primers used in this study are listed in Supplementary Table 4. Unless stated otherwise, polymerase chain reaction (PCR) was carried out using Q5 polymerase (NEB M0492L) for cloning purposes and GoTaq Green DNA polymerase (Promega M7123) for diagnostic purposes, both according to the manufacturer's instructions. Plasmid DNA and PCR fragments were purified using the PurePlasmid miniprep kit (CW Biosciences CW0500M) or the DNA Clean-up kit (CW Biosciences CW2301M), respectively. Details on strain and plasmid construction can be found in the Supplementary Information. No statistical methods were used to predetermine sample sizes used for experiments, but sample sizes are in line with field standards.

## Isolation of toxic *murA* alleles

Alleles of *murA* that are toxic when overexpressed were isolated as described previously[36] with slight modifications. First, the *murA* gene was mutagenized by PCR with Taq polymerase (NEB M0267L) from purified pKH37 plasmid using primer pair 15/76 in five separate reactions. A 500 ng quantity of each of the five resulting *murA** pools was then separately used as a megaprimer to amplify 50 ng of pKH37 using Q5 polymerase (NEB M0492L) in 50-µl reactions with the following thermocycler settings: 1) 95 °C for 3 min; 2) 95 °C for 50 s; 3) 60 °C for 50 s; 4) 72 °C for 10 min; 5) repeat steps 2–4 for a total of 25 cycles. Twenty units of DpnI (NEB R0176L) was subsequently added to each reaction and digestion of unamplified DNA was allowed to proceed at 37 °C for 1.5 h. Each reaction was drop dialysed using 0.025-µm mixed cellulose ester membranes (Millipore VSWP02500) floated on Milli-Q water for 20 min. A 12 µl volume of each dialysed product was then separately transformed into 100 µl of NEB-5-alpha electrocompetent *E. coli* (NEB C2989K), and recovered in SOC outgrowth medium (NEB B9020S), and transformants were selected by plating the outgrowth onto LB agar supplemented with gentamycin. Approximately 1 million colonies from each of the five libraries were separately pooled and the pKH37 derivatives were purified using the PurePlasmid miniprep kit (CW Biosciences CW0500M).

PAO1 was grown in 25 ml LB overnight at 37 °C, centrifuged at 12,000*g* for 5 min, and resuspended in an equal volume of 300 mM sucrose. The centrifugation and resuspension steps were repeated for a total of four washes. After a final centrifugation at 12,000*g* for 5 min, the cell pellet was resuspended in 1/20 of the original volume. A 1.2 µg quantity of pKH37-derived libraries was separately transformed into 150 µl of electrocompetent PAO1. Transformation reactions were recovered in LB at 37 °C for 1 h and transformants were selected by plating the outgrowth onto LB agar supplemented with 30 µg ml⁻¹ gentamycin. Approximately 1–2 million colonies from each library were separately pooled.

To identify pKH37 variants that harbour a toxic allele of *murA*, each PAO1 × pKH37* library was diluted to an optical density at 600 nm (OD$_{600nm}$) of 0.01 in 3 ml LB supplemented with 30 µg ml⁻¹ gentamycin and grown at 37 °C for 2 h 45 min. IPTG was added to each culture to a final concentration of 1 mM and the incubation at 37 °C was continued for an additional 2 h. Each culture was then subjected to centrifugation at 21,130*g* for 5 min and released DNA from each library was separately purified from 2.5 ml of the supernatant using the DNA clean-up kit (CW Biosciences CW2301M). The purified DNA was subsequently re-transformed into PAO1 electrocompetent cells prepared as described above and transformants were selected by plating the outgrowth on LB agar supplemented with gentamycin. The resulting colonies were then patched onto LB agar plates supplemented with gentamycin with and without 1 mM IPTG and grown overnight at 37 °C. The *murA* allele encoded by IPTG-sensitive isolates was determined by Sanger sequencing using primer pair 15/76.

## Isolation of MurA(C117S) loss-of-toxicity variants

pKH100 was mutagenized by passaging it through *E. coli* strain XL1-red. pKH100 was transformed into XL1-red and five separate cultures were inoculated, each using four unique transformants. After propagation overnight, plasmids were purified using the PurePlasmid miniprep kit (CW Biosciences CW0500M) to create five pKH100* libraries. PAO1 was grown in 25 ml LB overnight at 37 °C, centrifuged at 12,000*g* for 5 min, and resuspended in an equal volume of 300 mM sucrose. The centrifugation and resuspension steps were repeated for a total of four washes. After a final centrifugation at 12,000*g* for 5 min, the cell pellet was resuspended in 1/20 of the original volume. A 1.2 µg quantity of each pKH100* was separately transformed into 150 µl of electrocompetent PAO1. Transformation reactions were recovered in LB at 37 °C for 1 h and transformants were selected by plating the outgrowth onto LB agar supplemented with gentamycin. Approximately 1–10 million colonies from each library were separately pooled and stored in LB + 10% dimethylsulfoxide (DMSO) at −80 °C.

PAO1 × pKH100* libraries were plated on LB agar supplemented with 30 µg ml⁻¹ gentamycin and 1 mM IPTG, which allowed growth of only approximately 0.3–1% of all colony-forming units. A total of 17 or 18 IPTG-resistant colonies from each pool were streak-purified and the expression level and molecular weight of F−MurA(C117S)* from each isolate were determined by western blot analysis after growth in LB and induction for 1 h in mid-log phase. Only those isolates with expression levels and molecular weights consistent with F−MurA(C117S) were subjected to Sanger sequencing with primers 15 and 76. In three of the five libraries analysed, *F−murA^{WT}* was recovered. *F−murA^{C117S/G58D}* and *F−murA^{C117S/E406K}* were each recovered in one of the five libraries.

## Protein purification

Details on protein purification can be found in the Supplementary Information.

## Viability assays

Overnight cultures grown at 30 °C or 37 °C were centrifuged at 12,000*g* for 2 min and resuspended to an OD$_{600nm}$ of 1 in LB. Resuspensions were serially diluted in LB to 10⁻⁶ and 5 µl of each dilution was spotted onto LB agar supplemented with the appropriate inducers and antibiotics. Plates were incubated at 37 °C overnight and imaged using a Nikon D3400 camera equipped with a Nikon AF-S micro NIKKOR 40-mm lens.

## Western blot analysis

Overnight cultures were diluted to an OD$_{600nm}$ of 0.01 in 3 ml LB, grown at 37 °C for 4 h, and 2 ml was subjected to centrifugation at 12,000*g* for 3 min. When necessary, IPTG was added to 1 mM or arabinose was added to 0.1% 1 h before collecting the cultures. The cell pellet was resuspended to an OD$_{600nm}$ of 20 in 1× SDS sample buffer (50 mM Tris pH 6.8, 10% glycerol, 2% SDS, 0.2% bromophenol blue, 1% β-mercaptoethanol) and boiled at 100 °C for 10 min. The lysate was subsequently sonicated using a Qsonica Q800R3 sonicator with 30 cycles of 1 s on, 1 s off at 25% amplitude. Protein concentration was quantified using the Non-Interfering protein assay kit (G-biosciences 786-005). A 10 µg quantity of protein was loaded onto a 10% polyacrylamide−SDS gel and subjected to electrophoresis at 150 V for 70 min. Protein was transferred to PVDF membranes using the mixed molecular weight protocol on a Bio-Rad Trans-Blot Turbo transfer system and membranes were blocked in TBST (20 mM Tris pH 8.0, 200 mM NaCl, 0.05%

Tween-20) with 5% milk at room temperature for 1 h. Membranes were then probed with anti-His (GenScript A00186-100), anti-Flag (Sigma F7425) or anti-RpoA (BioLegend 663104) diluted 1:3,000, 1:3,000 or 1:100,000, respectively, in TBST + 5% milk at room temperature for 1 h. Membranes were washed three times for 5 min each in TBST and probed with peroxidase-conjugated goat-anti-mouse (Rockland 610-1302) or rabbit TrueBlot: anti-rabbit IgG HRP (Rockland 18-8816-33) diluted 1:3,000 in TBST + 5% milk at room temperature for 1 h. Membranes were washed three times for 5 min each in TBST, developed with SuperSignal West Pico PLUS Chemiluminescent Substrate (Thermo 34580), and imaged using an Azure Biosystems C600 imager.

## In vivo co-affinity purification

To identify in vivo interacting partners of *Pa*LpxC in an unbiased manner, two co-affinity purification schemes were used in replicate samples. Specifically, in condition 1, 150 mM NaCl was used in lysis and wash buffers, but in condition 2, 300 mM NaCl was used in lysis and wash buffers. Overnight *P. aeruginosa* cultures were diluted to $OD_{600nm} = 0.01$ in 500 ml LB and grown at 37 °C to early logarithmic phase. In the case of PA1009, the culture was induced with 0.5% arabinose 1 h before collection. The entire culture was subjected to centrifugation at 13,000$g$ for 10 min and the pellets were resuspended in 4 ml of lysis buffer (50 mM Tris pH 7.5, 2% glycerol, 5 mM imidazole, 1% Triton X-100, and 150 mM or 300 mM NaCl) and cells were lysed by sonication with a Q125 Qsonica sonicator. Cell debris was pelleted by centrifugation at 21,130$g$ for 15 min at 4 °C and the clarified supernatant was mixed with Ni-NTA resin (Qiagen 30230) rotating end-over-end at 4 °C for 1.5 h. Resin was washed four times with 6 ml of wash buffer (50 mM Tris pH 7.5, 2% glycerol, 25 mM imidazole, 1% Triton X-100, and 150 mM or 300 mM NaCl). Proteins were eluted from the resin in elution buffer (50 mM Tris pH 7.5, 2% glycerol, 250 mM imidazole, 0.1% Triton X-100 and 150 mM NaCl). Eluates were mixed 1:1 with 2× SDS sample buffer (100 mM Tris pH 6.8, 20% glycerol, 4% SDS, 0.4% bromophenol blue), and 40 µl was loaded onto 4–20% polyacrylamide Mini-PROTEAN TGX gels (Bio-Rad 4568094) and subjected to electrophoresis at 80 V for 25 min. The gel was subsequently stained with Coomassie R-250 and entire lanes were excised for mass spectrometry analysis.

Excised lanes were cut into approximately 1-mm³ pieces. Gel pieces were then subjected to a modified in-gel trypsin digestion procedure[37]. Gel pieces were washed and dehydrated with acetonitrile for 10 min, followed by removal of acetonitrile. Pieces were then completely dried in a speed-vac. Gel pieces were rehydrated with 50 mM ammonium bicarbonate solution containing 12.5 ng µl⁻¹ modified sequencing-grade trypsin (Promega) at 4 °C. After 45 min, the excess trypsin solution was removed and replaced with 50 mM ammonium bicarbonate solution to just cover the gel pieces. Samples were then placed in a 37 °C room overnight. Peptides were later extracted by removing the ammonium bicarbonate solution, followed by one wash with a solution containing 50% acetonitrile and 1% formic acid. The extracts were then dried in a speed-vac (about 1 h). The samples were then stored at 4 °C until analysis.

On the day of analysis, the samples were reconstituted in 5–10 µl of high-performance liquid chromatography (HPLC) solvent A (2.5% acetonitrile, 0.1% formic acid). A nanoscale reversed-phase HPLC capillary column was created by packing 2.6-µm C18 spherical silica beads into a fused silica capillary (100 µm inner diameter × about 30 cm length) with a flame-drawn tip[38]. After equilibrating the column, each sample was loaded using a Famos auto sampler (LC Packings) onto the column. A gradient was formed, and peptides were eluted with increasing concentrations of solvent B (97.5% acetonitrile, 0.1% formic acid).

As peptides eluted, they were subjected to electrospray ionization and then entered into an LTQ Orbitrap Velos Pro ion-trap mass spectrometer (Thermo Fisher Scientific). Peptides were detected, isolated and fragmented to produce a tandem mass spectrum of specific fragment ions for each peptide. Peptide sequences (and hence protein identity) were determined by matching protein databases with the acquired fragmentation pattern by the software program Sequest v28 (rev. 13; Thermo Fisher Scientific)[39]. All databases include a reversed version of all the sequences, and the data were filtered to a peptide false-discovery rate of 1–2%. Only proteins that exhibited the following criteria were reported: at least five unique peptides detected in both PA1018 samples; and at least threefold enrichment in both PA1018 samples compared to the corresponding PAO1 and PA1009 samples.

## In vivo targeted pulldowns

Overnight cultures of *P. aeruginosa* strains encoding combinations of H−*Pa*LpxC, F−*Pa*MurA(WT) and F−*Pa*MurA(C117S) were diluted to $OD_{600nm} = 0.01$ in 50 ml LB and grown at 37 °C for 2.5 h. IPTG was added to each culture to a final concentration of 1 mM and cultures were grown for an additional 30 min at 37 °C before the addition of DMSO to 0.1% and CHIR-090 to 0.5 µg ml⁻¹ where indicated. Cultures were incubated at 37 °C for an additional hour and collected by centrifugation at 12,000$g$ for 10 min. Cells were washed in 1 ml LB and pelleted by centrifugation at 12,000$g$ for 2 min. Cells were resuspended in 1 ml lysis buffer (25 mM Tris pH 8.0, 150 mM NaCl, 2% glycerol, 0.1% Triton X-100, 5 mM imidazole, 0.1% DMSO with or without 0.5 µg ml⁻¹ CHIR-090 as indicated) and lysed by sonication on ice at 40% amplitude with four cycles of 15 s on, 15 s off. The extracts were clarified by two consecutive centrifugation steps at 21,130$g$ for 5 min at 4 °C and protein concentration was determined with the Non-Interfering protein assay kit (G-biosciences 786-005). Supernatants were diluted to 3.5 mg ml⁻¹ protein in 500 µl lysis buffer and mixed with 25 µl packed Ni-NTA resin (Qiagen 30230). Mixtures were incubated at 4 °C for 3 h rotating end-over-end. Beads were washed three times with 500 µl wash buffer (25 mM Tris pH 8.0, 150 mM NaCl, 2% glycerol, 0.1% Triton X-100, 25 mM imidazole, 0.1% DMSO with or without 0.5 µg ml⁻¹ CHIR-090 as indicated). Protein was eluted in 40 µl elution buffer (25 mM Tris pH 8.0, 150 mM NaCl, 2% glycerol, 0.1% Triton X-100, 500 mM imidazole).

A 10 µg quantity of protein per input sample and 10 µl of eluate samples were resolved by SDS−PAGE in 10% polyacrylamide gels. Protein was transferred to PVDF membranes using the mixed molecular weight protocol on a Bio-Rad Trans-Blot Turbo transfer system and membranes were blocked in TBST + 5% milk at room temperature for 1 h. Membranes were then probed with anti-His (GenScript A00186-100) or anti-Flag (Sigma-Aldrich F7425) diluted 1:3,000 in TBST + 5% milk at room temperature for 1 h. Membranes were washed three times for 5 min each in TBST and probed with peroxidase-conjugated goat-anti-mouse (Rockland 610-1302) or rabbit TrueBlot: anti-rabbit IgG HRP (Rockland 18-8816-33) diluted 1:3,000 in TBST + 5% milk at room temperature for 1 h. Membranes were washed three times for 5 min each in TBST, developed with SuperSignal West Pico PLUS Chemiluminescent Substrate (Thermo 34580), and imaged using an Azure Biosystems C600 imager.

## In vitro co-affinity purification

Purified F−*Pa*LpxC and H−MurA variants were each diluted to 2.5 µM in co-purification buffer (25 mM Tris pH 8.0, 150 mM NaCl, 10% glycerol, 1 mM dithiothreitol (DTT), 0.5% DMSO) to a final volume of 100 µl. When indicated, CHIR-090 was added to 5.7 µM. Mixtures were incubated on ice for 30 min after which 85 µl of each mixture was added to an equal volume of Anti-Flag M2 Magnetic Beads (Sigma-Aldrich M8823) and incubated at 4 °C for 1 h rotating end-over-end. Beads were collected with a magnetic rack and washed twice with 300 µl co-purification buffer and once with 200 µl co-purification buffer. When indicated, 5.7 µM CHIR-090 was maintained in the wash buffers. Proteins were eluted with 85 µl elution buffer (25 mM Tris pH 8.0, 150 mM NaCl, 10% glycerol, 1 mM DTT, 100 µg µl⁻¹ Flag peptide (Millipore Sigma F3290)) by incubation at room temperature for 30 min rotating end-over-end. Input samples (5 µl) and eluate samples (15 µl) were resolved by SDS−PAGE in 4–20% polyacrylamide Mini-PROTEAN TGX gels (Bio-Rad 4561095) and protein was detected with Coomassie staining.

## SEC

F–$Pa$LpxC and H–$Pa$MurA(WT) either individually or in combination were diluted to 7.5 μM each in 500 μl of SEC buffer 1 (10% glycerol, 0.33% DMSO, 1 mM DTT, 25 mM Tris pH 8.0 and 150 mM NaCl). When indicated, CHIR-090 was added to 37.5 μM. The mixtures were subsequently subjected to SEC using an ActaPure system equipped with a Superdex 200 10/300 GL column equilibrated with SEC buffer 2 (10% glycerol, 1 mM DTT, 25 mM Tris pH 8.0 and 150 mM NaCl) and eluate fractions were collected every 0.35 ml. For F–$Pa$LpxC and H–$Pa$MurA(WT) mixtures, three fractions corresponding to the shifted peak (14.1–15.2 ml with no CHIR-090, 13.8–14.8 ml for with CHIR-090) were pooled and concentrated by centrifugation at 4 °C using centrifugal filters with a molecular weight cutoff of 10-kDa (Amicon UFC801024). The concentrated protein was resubjected to SEC on a Superdex 200 10/300 GL column as described above. A 15 μl volume of the relevant fractions was resolved on a 4–20% polyacrylamide Mini-PROTEAN TGX gels (Bio-Rad 4568094) and protein was detected with Coomassie staining.

## Growth curves

Cultures grown overnight at 30 °C were pelleted at 12,000$g$ for 2 min and resuspended in fresh LB. Cultures were diluted in 96-well microtitre plates to an $OD_{600nm}$ of 0.01 in 200 μl LB. Cultures were grown in a Tecan Infinite M-Plex plate reader at 37 °C and $OD_{600nm}$ was monitored every 4 min with shaking at a 1-mm orbital for 200 s after each time point.

## LPS silver stain

Overnight cultures were diluted to an $OD_{600nm}$ of 0.01 in 25 ml LB cultures and grown for 4 h at 37 °C. For MurA- and MurB-depletion strains grown in the presence of inducer, 1 mM IPTG was present during the entire outgrowth. For MurA and LpxC overexpression strains, 1 mM IPTG or 0.1% arabinose was added after 3 h of outgrowth and incubation was continued for one additional hour. A 20 ml volume of each culture was centrifuged at 12,000$g$ for 5 min and the pellet was washed in 1 ml of LB before centrifugation at 12,000$g$ for 2 min. Pellets were resuspended to an $OD_{600nm}$ of 20 in LDS sample buffer (Invitrogen NP0008) + 4% β-mercaptoethanol and boiled at 100 °C for 10 min. Samples were sonicated with a Q125 Qsonica sonicator at 25% amplitude for 1 s on, 1 s off for 30 cycles and protein concentration was determined using the Non-Interfering protein assay kit (G-biosciences 786-005). A 5 μg quantity of protein was loaded onto a 15% polyacrylamide–SDS gel and subjected to electrophoresis at 150 V for 70 min before western blot analysis of RpoA as described above. For the LPS gel, 50 μl of each sample was mixed with 1.25 μl of proteinase K (NEB P8107S) and incubated at 55 °C for 1 h followed by 95 °C for 10 min. The equivalent volume accounting for 10 μg of protein was resolved on a 4–12% Criterion XT Bis-Tris gel (Bio-Rad 3450124) at 100 V for 1 h 45 min and LPS was detected by silver stain as described previously[40]. Briefly, the gel was fixed by incubation in 200 ml of 40% ethanol, 5% acetic acid overnight. Periodic acid was added to 0.7% and allowed to incubate for 5 min. The gel was subsequently washed with three changes of 200 ml ultrapure water over the course of 2 h. The gel was impregnated with 150 ml of freshly made staining solution (0.018 N NaOH, 0.4% NH₄OH and 0.667% silver nitrate) for 10 min and subsequently washed with three changes of 200 ml ultrapure water over the course of 2 h. The gel was developed with 200 ml freshly prepared developer solution (0.26 mM citric acid pH 3.0, 0.014% formaldehyde) and development stopped in 100 ml of 1% acetic acid. The gel was imaged in a Bio-Rad ChemiDoc MP Imaging System.

## Microscopy

Overnight cultures were diluted to an $OD_{600nm}$ of 0.01 in 3 ml LB with 1 mM IPTG as appropriate. Cultures were grown at 37 °C for 2.5 h and, when appropriate, antibiotics were added to the following concentration: 0.5 μg ml⁻¹ CHIR-090, 64 μg ml⁻¹ fosfomycin or 0.5 μg ml⁻¹ CHIR-090 + 16 μg ml⁻¹ fosfomycin. Cultures were grown at 37 °C for an additional 1.5 h and 1 ml was pelleted at 12,000$g$ for 2 min. Cell pellets were resuspended to an $OD_{600nm}$ of 10 in LB, spotted onto an LB + 1.5% agar pad, and imaged on a Nikon Eclipse 50i microscope equipped with a 100× objective and a Photometrics CoolSNAP HQ² camera. Images were uniformly edited using the Fiji image analysis platform version 2.3.0/1.53q[41]. Cell width was quantified using the MicrobeJ plugin (version 5.13I) of ImageJ using the following settings: area: 250–max, length: 3–100, width: 5–25, width variation: 0–0.25, width symmetry: 0–1, circularity, 0.01–1. Normality of each population was tested and determined to be lacking by both the D'Agostino and Pearson test and the Anderson–Darling test carried out by Prism (GraphPad Software, LLC). Statistical significance was determined by Kruskal–Wallis test carried out by Prism (GraphPad Software, LLC). Microscopy source images are available at https://doi.org/10.5281/zenodo.7455522 (ref. [42]).

## Determination of LpxC product and substrate levels in vivo

Overnight cultures were diluted to an $OD_{600nm}$ of 0.01 in 200 ml LB, grown at 37 °C for 3 h, and IPTG was added to a final concentration of 1 mM. After incubation at 37 °C for one additional hour, cells were collected by centrifugation at 12,000$g$ for 5 min. After resuspension in LB, cells were re-pelleted by centrifugation at 12,000$g$ for 2 min. Pellets were then resuspended in 400 μl HPLC-grade $H_2O$. Subsequently, 250 μl chloroform and 500 μl methanol were added to the mixture. Samples were vortexed vigorously, placed on ice for 10 min, and then spun down at 4 °C at 21,000$g$. A 500 μl volume of the aqueous layer was collected and dried down at 30 °C overnight in a rotary evaporator. Dried material was resuspended in 100 μl ultrapure $H_2O$ and the $Pa$LpxC product and substrate were detected by liquid chromatography–tandem mass spectrometry (LC–MS/MS) as described below.

## In vitro LpxC activity assay

Purified F–$Pa$LpxC and H–MurA variants were diluted to 200 nM in binding buffer (25 mM Tris pH 8.0, 150 mM NaCl, 6% glycerol) to a final volume of 15 μl and mixtures were allowed to equilibrate to 30 °C for 10 min. The reaction was started by the addition of 15 μl of substrate mixture (25 mM Tris pH 8.0, 150 mM NaCl, 4% DMSO, 200 μM UDP-3-$O$-($R$-3-hydroxydecanoyl)-GlcNAc (Carbosynth MU75071)) and allowed to proceed for 5 or 10 min at 30 °C at which point the reaction was stopped by the addition of 45 μl 2% acetic acid. Stopped reactions were frozen on dry ice, lyophilized, and the lyophilized material was resuspended in 60 μl $H_2O$. $Pa$LpxC product and substrate were detected by LC–MS as described below. The linear range of the reaction was determined as described above with purified F–$Pa$LpxC alone and reactions were stopped after 1, 2, 5, 10, 15 or 20 min (Extended Data Fig. 5b). H–$Pa$MurA(WT) and H–$Pa$MurA(C117S) alone exhibited background levels of LpxC activity, confirming that the observed stimulation of LpxC activity was not due to contamination within the protein preparation or an alternative enzymatic activity of $Pa$MurA (Extended Data Fig. 6e).

Activity assays with the $L. pneumophila$ and $E. coli$ LpxC–MurA pairs were carried out as described above except that concentration of H–MurA variants in the final reaction mixture was 200 nM. $L. pneumophila$ reactions were stopped after 10 min and $E. coli$ reactions were stopped after 1 min.

## LC–MS

To quantify the activity of LpxC in vitro, the $Pa$LpxC product and substrate were detected by LC–MS on an Agilent Technologies 1200 series HPLC system in line with an Agilent 6520 Q-TOF mass spectrometer with electrospray ionization–mass spectrometry operating in negative mode. Samples were separated on a Waters Symmetry C18 column (5 μm; 3.9 mm × 150 mm) with a matching column guard using the following method: flow rate = 0.5 ml min⁻¹, 95% solvent A ($H_2O$, 10 mM ammonium formate) and 5% solvent B (acetonitrile) for 5 min

followed by a linear gradient of solvent B from 5–80% over 20 min. Agilent MassHunter Workstation Qualitative Analysis software version B.06.00 was used for analysing the MS data. The $Pa$LpxC substrate (UDP-3-$O$-($R$-3-hydroxydecanoyl)-$N$-acetylglucosamine) was quantified by monitoring the abundance of 776.21 $m/z$ and resolved as a single peak, which was integrated to infer substrate concentration. The $Pa$LpxC product (UDP-3-$O$-($R$-3-hydroxydecanoyl)-glucosamine) was quantified by monitoring the abundance of 734.1944 $m/z$ and often resolved as multiple peaks as reported previously[43], all of which were integrated to infer product concentration.

$Pa$LpxC substrate and product from the aqueous fraction of methanol–chloroform-extracted whole-cell lysates were analysed by LC–MS/MS using the same settings as the LC–MS analysis described above with the following adaptations. Parent ions with $m/z$ of 776.1986 (corresponding to the $Pa$LpxC substrate) and $m/z$ of 734.1872 (corresponding to the $Pa$LpxC product) were targeted for MS/MS with a collision energy of 40. Fragment ions between 50 and 850 $m/z$ were analysed. The relative abundance of the $Pa$LpxC substrate and product were quantified by integrating the peaks observed for 776.1986 $m/z$ and 734.1872 $m/z$, respectively. Raw LC–MS/MS data are available at https://doi.org/10.5281/zenodo.7455522 (ref. [42]).

### In vitro MurA activity assay

Purified MurA variants were diluted to 200 nM (for $Pa$MurA(WT), $Pa$MurA(G58D) and $Pa$MurA(E406K)) or 2 µM (for $Pa$MurA* variants) in MurA reaction buffer (25 mM Tris pH 7.75, 6% glycerol) to a final volume of 25 µl and allowed to equilibrate to room temperature for 30 min. The reaction was started by the addition of 25 µl of MurA substrate mixture (25 mM Tris pH 7.75, 2 mM UDP-$N$-acetylglucosamine (Sigma-Aldrich U4375), and 1 mM phosphoenolpyruvate (Sigma-Aldrich 10108294001)) and the reaction was allowed to proceed for 30 min at 37 °C. The reaction was stopped by the addition of 800 µl Lanzetta reagent (0.033% malachite green, 1% ammonium molybdate, 1 N HCl, 0.2% Triton X-100)[44] and was incubated at room temperature for 1.25 min before the addition of 100 µl 34% sodium citrate. After an additional 30-min incubation at room temperature, the absorbance at 600 nm ($A_{660nm}$) of each sample was determined. A standard curve made from NaH$_2$PO$_4$ diluted in ultrapure water was used to quantify the amount of free P$_i$ released by the reaction.

### MurA–LpxC complex structure prediction

The structure of the LpxC–MurA complex was predicted using the default parameters of AlphaFold[19]. The $P. aeruginosa$ PAO1 LpxC and MurA sequences or the $E. coli$ MG1655 LpxC and MurA sequences were separated by a linker containing 15 repeats of Gly-Gly-Gly-Ser, which was not displayed for clarity. The structure was visualized using ChimeraX version 1.1.1 (ref. [45]).

### Predicting complex interactions

We sought to use evolutionary covariation to predict the organisms in which the LpxC–MurA interaction does or does not exist. We used EVComplex run with EVcouplings v0.0.5 (ref. [18]) informed by the AlphaFold model to define an interaction score based on the inferred pairwise interaction terms $J_{ij}$ of the complex and validate our proposed complex structure.

We first generated multiple sequence alignments of LpxC and MurA using jackhammer v3.1b2 (ref. [46]) with five iterations on the Uniref100 dataset[47] released in February 2021. Both $P. aeruginosa$ and $E. coli$ MurA (Uniprot: Q9HVW7 and P0A749) and LpxC (Uniprot: P47205 and P0A725) were used as initial sequences. In each case, alignments were constructed with a range of sequence score thresholds from 0.1 to 0.8 bits per residue, which did not meaningfully change which sequences were included for either species or protein. The same sequences were collected when starting from either species. For all analyses, we began with alignments of 54,940 MurA and 23,634 LpxC sequences generated

from a relative bitscore of 0.8, and then concatenated MurA and LpxC sequences for each species, resolving ambiguities in pairing by selecting the sequence in each species closest to the original $P. aeruginosa$ homologue. The final concatenated alignment contained 11,834 sequences. From this concatenated multiple sequence alignment, evolutionary couplings were determined using pseudo-likelihood maximization[48].

We more closely analysed the top five ranked intermolecular couplings ($J_{ij}$) of the direct coupling analysis model that correspond to pairs of amino acid positions in structural contact in the AlphaFold2 model, defined as having Cα atoms within 9 Å. Each pair of sequences can be scored on the coupling between MurA position $i$ and LpxC position $j$ on the basis of the appropriate $J_{ij}$ term from the model. Summing these five scores for each species, we observed a bimodal distribution (Extended Data Fig. 9c). We selected only pairs of sequences whose sum over the five scores exceeded 0.1, totalling 1,689 sequence pairs (Extended Data Fig. 9c), and trained a new EVcomplex model on just these sequences from the concatenated alignment. In this model, five of the top six intermolecular couplings corresponded to predicted contacts in the AlphaFold2 structure (Extended Data Fig. 9d). We defined a final 'interaction score' based on the top 20 coupling terms from this refined model and used it to score all 11,834 pairs of sequences in the concatenated alignment again. The sequence analysis files are available at https://doi.org/10.5281/zenodo.7455522 (ref. [42]). The top 20 terms were chosen arbitrarily, but this exact choice does not meaningfully affect downstream analyses, and all conclusions about relative interaction scores remain true for choices of top couplings between five and five hundred.

To estimate the percentage of interacting sequences in each clade, we fitted a two-component Gaussian mixture model using sklearn v1.0.2 (ref. [49]) to the interaction scores for all sequence pairs and calculated the posterior probability of each sequence being drawn from the upper distribution. We mapped the taxonomic identifier for each sequence pair to its class in the National Center for Biotechnology Information taxonomy database using ETE3 v1.3.2 (ref. [50]), identifying 2,459 alphaproteobacteria and 2,781 gammaproteobacteria in the alignment. We then estimated the expected fraction of interacting sequences in each clade as the average of the interaction probabilities of each sequence in that clade.

To visualize the phylogenetic structure of these sequences, we built a maximum-likelihood phylogeny of MurA and LpxC from the concatenated multiple sequence alignment with FastTree version 2.1.11 (refs. [51]). Interaction scores were mapped onto the tree in Python v3.8.8 and visualized with the interactive tree of life (ITOL) v5 (ref. [52]). Custom code used in this study can be found at https://github.com/samberry19/evcomplex-interaction-scoring or https://doi.org/10.5281/zenodo.7471436 (ref. [53]).

### Reporting summary

Further information on research design is available in the Nature Portfolio Reporting Summary linked to this article.

### Data availability

LC–MS/MS source data, microscopy images and sequence analysis files used to derive LpxC–MurA interaction scores are available at https://doi.org/10.5281/zenodo.7455522. Uniprot accession codes for genes used to generate LpxC–MurA interaction scores are included in the Methods and source data. All bacterial strains and plasmids developed in this study are available upon request. Source data are provided with this paper.

### Code availability

All code generated in this study is available at https://github.com/samberry19/evcomplex-interaction-scoring or https://doi.org/10.5281/zenodo.7471436.

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

**Acknowledgements** We thank all of the members of the Bernhardt, Rudner and Mekalanos labs for their thoughtful discussions and advice throughout this project, especially L. Marmont for technical insight regarding protein purification. We also thank S. Lory for strains and reagents as well as advice on the genetic manipulation of *P. aeruginosa*, B. Silva and M. Welsh for assistance with LC–MS/MS analysis, and the Taplin Mass Spectrometry Core Facility at Harvard Medical School for the identification of co-affinity-purified proteins. LC–MS data were acquired on an Agilent 6520 Q-TOF spectrometer supported by the Taplin Funds for Discovery Program. Protein structures were visualized with UCSF ChimeraX, developed by the Resource for Biocomputing, Visualization, and Informatics at the University of California, San Francisco, with support from National Institutes of Health R01-GM129325 and the Office of Cyber Infrastructure and Computational Biology, National Institute of Allergy and Infectious Diseases. This work was supported by the National Institutes of Health (F32AI164630 to K.R.H., AI148752 to S.W., AI083365 to T.G.B., and U19 AI158028 to T.G.B and S.W.), the Chan Zuckerberg Initiative (CZI2018-191853 to D.S.M.) and Investigator funds from the Howard Hughes Medical Institute (T.G.B).

**Author contributions** K.R.H. and T.G.B. designed the study and wrote the manuscript. S.P.B., J.K.M. and D.S.M. devised and carried out the covariation analysis. Z.L. established the methanol–chloroform metabolite extraction protocol. A.T. and S.W. developed the LC–MS protocol. K.R.H. carried out all other experiments and data analysis. All authors read and approved the manuscript.

**Competing interests** The authors declare no competing interests.

**Additional information**
**Correspondence and requests for materials** should be addressed to Thomas G. Bernhardt.

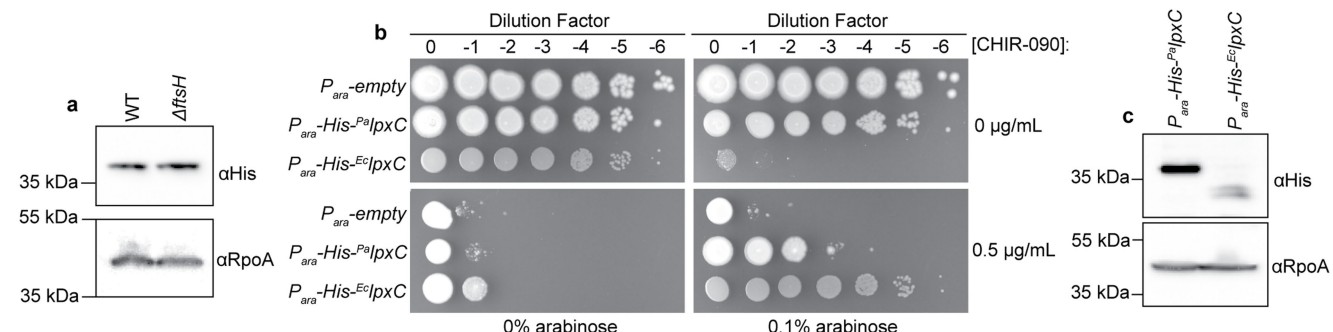

**Extended Data Fig. 1 | Regulation of LpxC in *P. aeruginosa* differs from that observed in *E. coli*.** (a) Anti-His immunoblot detecting H-*Pa*LpxC expressed from the native chromosomal locus in wild-type cells or an *ftsH* deletion mutant. A corresponding blot for RpoA was used as a loading control. Data are representative of 3 biological replicates. (b) Spot titer assay in which serial dilutions of PAO1 harboring an empty plasmid or one encoding *His-Pa lpxC* or *His-Ec lpxC* under arabinose-inducible control were plated on LB agar supplemented with arabinose and/or the LpxC inhibitor CHIR-090 as indicated. Plates were incubated at 37 °C for 20 h before being imaged. Data are representative of 3 biological replicates. (c) Anti-His immunoblot analysis of His-*Pa*LpxC or His-*Ec*LpxC protein levels in exponentially growing PAO1. Immunoblot for RpoA serves as a loading control. Data are representative of 3 biological replicates. For gel source data, see Supplementary Fig. 1.

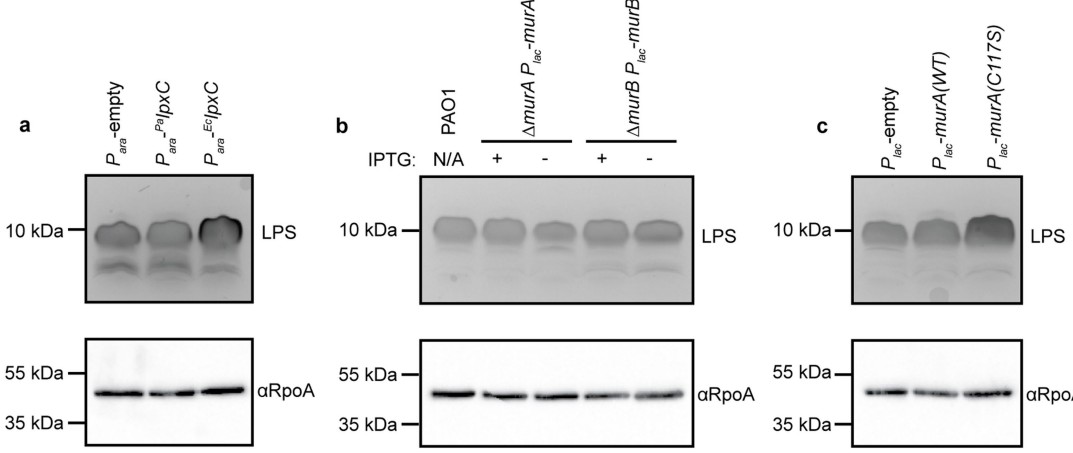

**Extended Data Fig. 2 | LPS levels are altered upon mis-regulation of LpxC.**
Silver stain of LPS harvested from exponentially growing cultures and western blot of RpoA from the same samples as a loading control. (a) PAO1 harboring an empty plasmid or one encoding $^{Pa}lpxC$ or $^{Ec}lpxC$ under arabinose-inducible control were induced with 0.1% arabinose 1 h prior to harvesting samples. (b) PAO1, PA1118 [$\Delta murA\ P_{lac}$-$murA$], or PA1135 [$\Delta murB\ P_{lac}$-$murB$] were grown in the presence or absence of 1 mM IPTG as indicated before samples were processed. (C) PAO1 harboring an empty plasmid or one encoding $^{Pa}murA(WT)$ or $^{Pa}murA(C117S)$ under IPTG-inducible control were induced with 1 mM IPTG 1 h prior to harvesting samples. Data are representative of 3 biological replicates. For gel source data, see Supplementary Fig. 1.

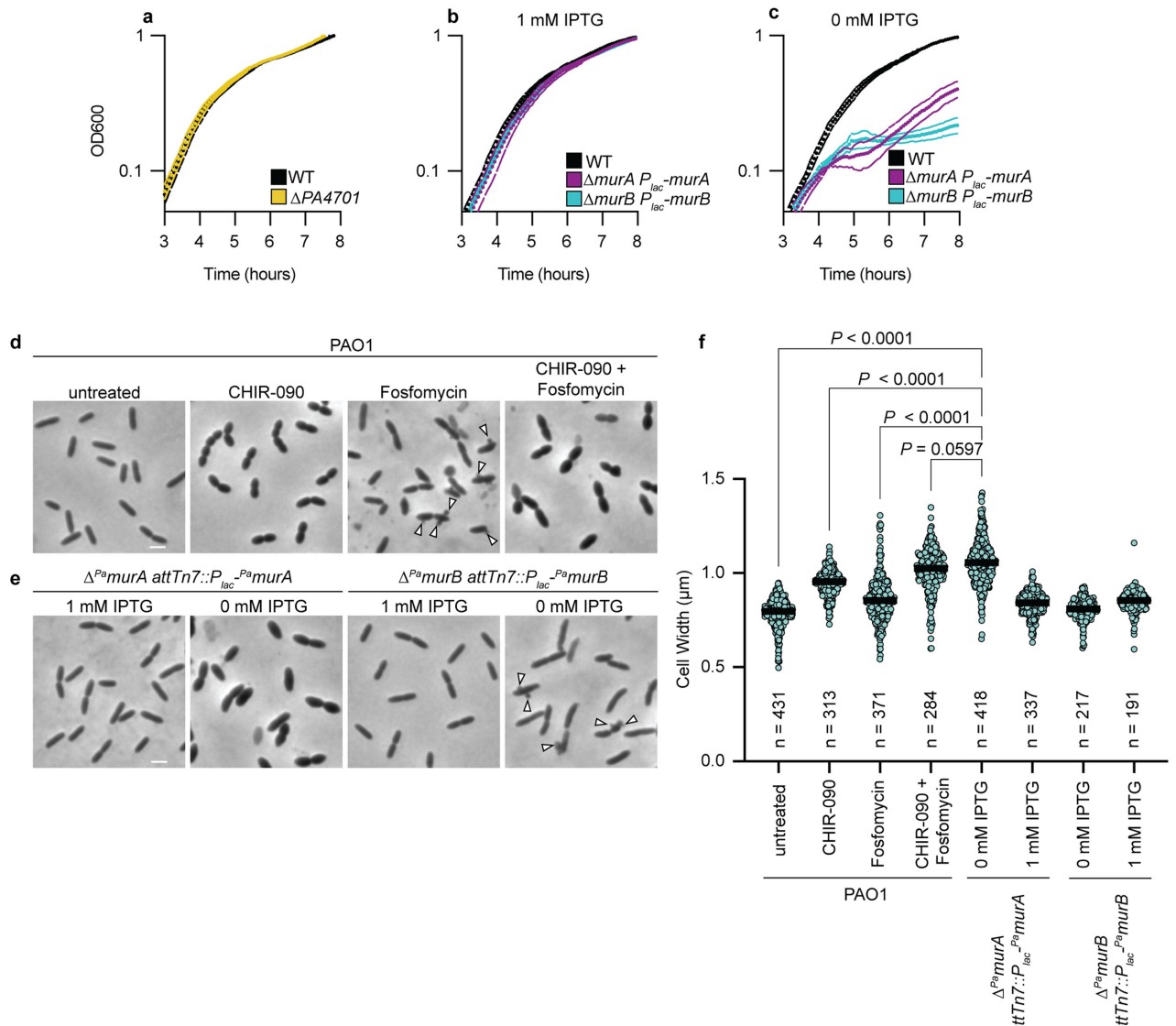

**Extended Data Fig. 3 | MurA is essential and its depletion phenocopies simultaneous inhibition of PG and LPS biosynthesis.** (a–c) Growth curves of *P. aeruginosa* strains in LB with or without IPTG as indicated. Dots represent the average of 3 biological replicates and dashed lines indicate the standard deviation. The following strains were used: (a) PAO1 [WT] and PA1080 [*ΔPA4701*], (b,c) PAO1 [WT], PA1118 [*ΔmurA P_lac-murA*], and PA1135 [*ΔmurB P_lac-murB*]. (d) Phase contrast images of *P. aeruginosa* cells after 1 hr treatment with the indicated antibiotic(s). Scale bar indicates 2 μm. Data are representative of at least 2 biological replicates (e) Phase contrast images of the indicated *P. aeruginosa murA* or *murB* depletion strains grown for 4 h in the presence or absence of inducer as indicated. MurB depletion was analyzed as a control to compare the phenotype of inactivating another early step in PG synthesis with that of MurA. Scale bar indicates 2 μm. Data are representative of at least 3 biological replicates (f) Quantification of cell width after 1 hr treatment with the indicated antibiotic(s) or after depletion of MurA or MurB. Each dot represents an individual cell and the median of the population is indicated by a black line. n indicates the number of cells analyzed. For each condition, the cells quantified were derived from a single population and data are representative of biological duplicates.

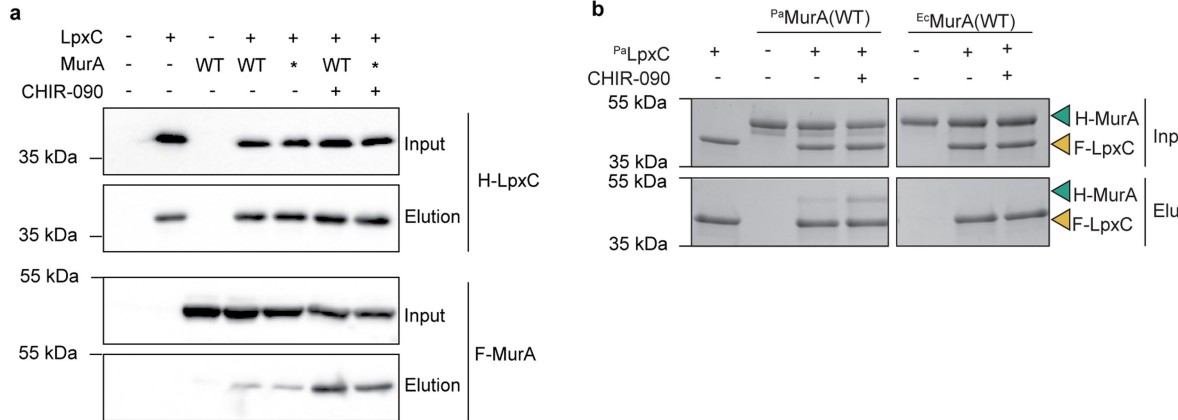

**Extended Data Fig. 4 | $^{Pa}$MurA interacts with $^{Pa}$LpxC.** (a) H-$^{Pa}$LpxC *in vivo* pulldowns. The expression status of H-$^{Pa}$LpxC and F-$^{Pa}$MurA before (input) and after (elution) co-affinity purification using Ni-NTA resin is indicated above the immunoblots. The variant of F-$^{Pa}$MurA produced is indicated by WT for F-$^{Pa}$MurA(WT) or * for F-$^{Pa}$MurA(C117S). When indicated, 0.5 μg/mL CHIR-090 was added to cultures 1 hr prior to harvesting and was maintained in all lysis and wash buffers as detailed in the methods section. The following strains were used to generate the lysates: PA239 (lane 1), PA1013 (lane 2), PA1068 (lane 3), PA1071 (lanes 4 and 6), and PA1121 (lanes 5 and 7). Data are representative of at 3 biological replicates. (b) Purified F-$^{Pa}$LpxC (2.5 μM) was mixed in a 1:1 ratio with purified H-MurA variants in the presence or absence of CHIR-090 (5.7 μM) as indicated. The mixtures were pulled down with anti-FLAG resin and the input and elution subjected to SDS-PAGE and Coomassie staining. Mobilities of H-MurA and F-LpxC are indicated by a cyan or gold carrot, respectively. Data are representative of 3 replicates. For gel source data, see Supplementary Fig. 1.

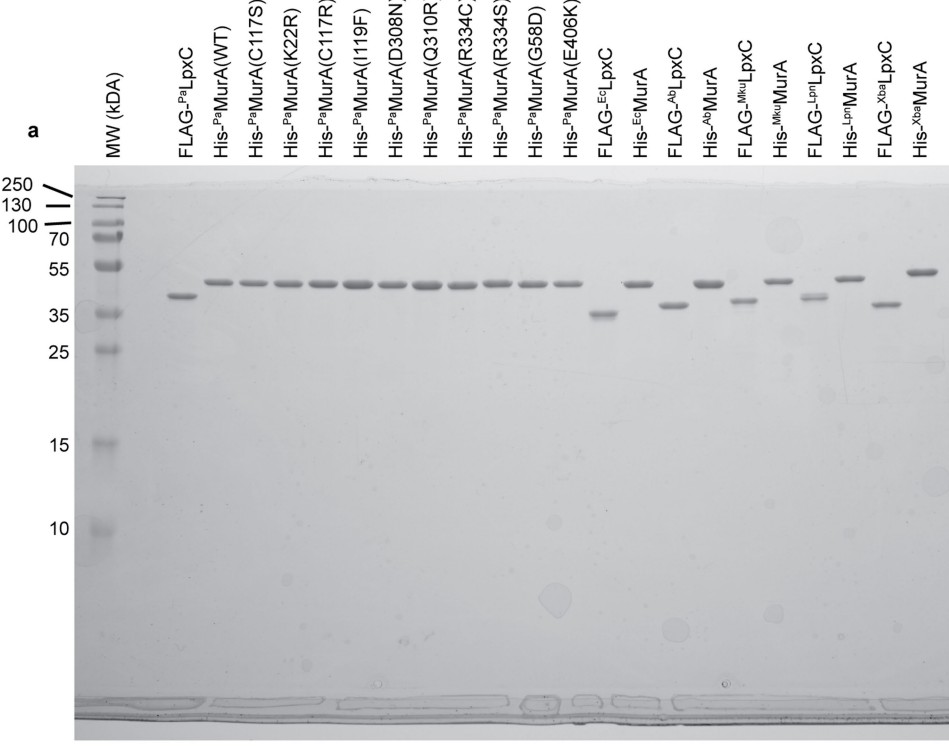

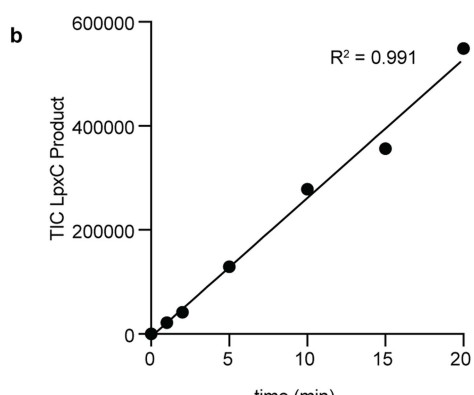

**Extended Data Fig. 5 | Purified proteins used in this study and linearity of LpxC activity assay.** (a) Purified proteins used in this study were resolved by SDS-PAGE and protein was visualized with Coomassie staining. Data are representative of at least two 2 replicates. For gel source data, see Supplementary Fig. 1. (b) Time course in which turnover of UDP-3-O-(R-3-hydroxydecanoyl)-N-acetylglucosamine ($^{Pa}$LpxC substrate) to UDP-3-O-(R-3-hydroxydecanoyl)-glucosamine ($^{Pa}$LpxC product) by FLAG-$^{Pa}$LpxC over the course of 20 min was monitored by LC-MS. The $R^2$ value of the linear regression is presented.

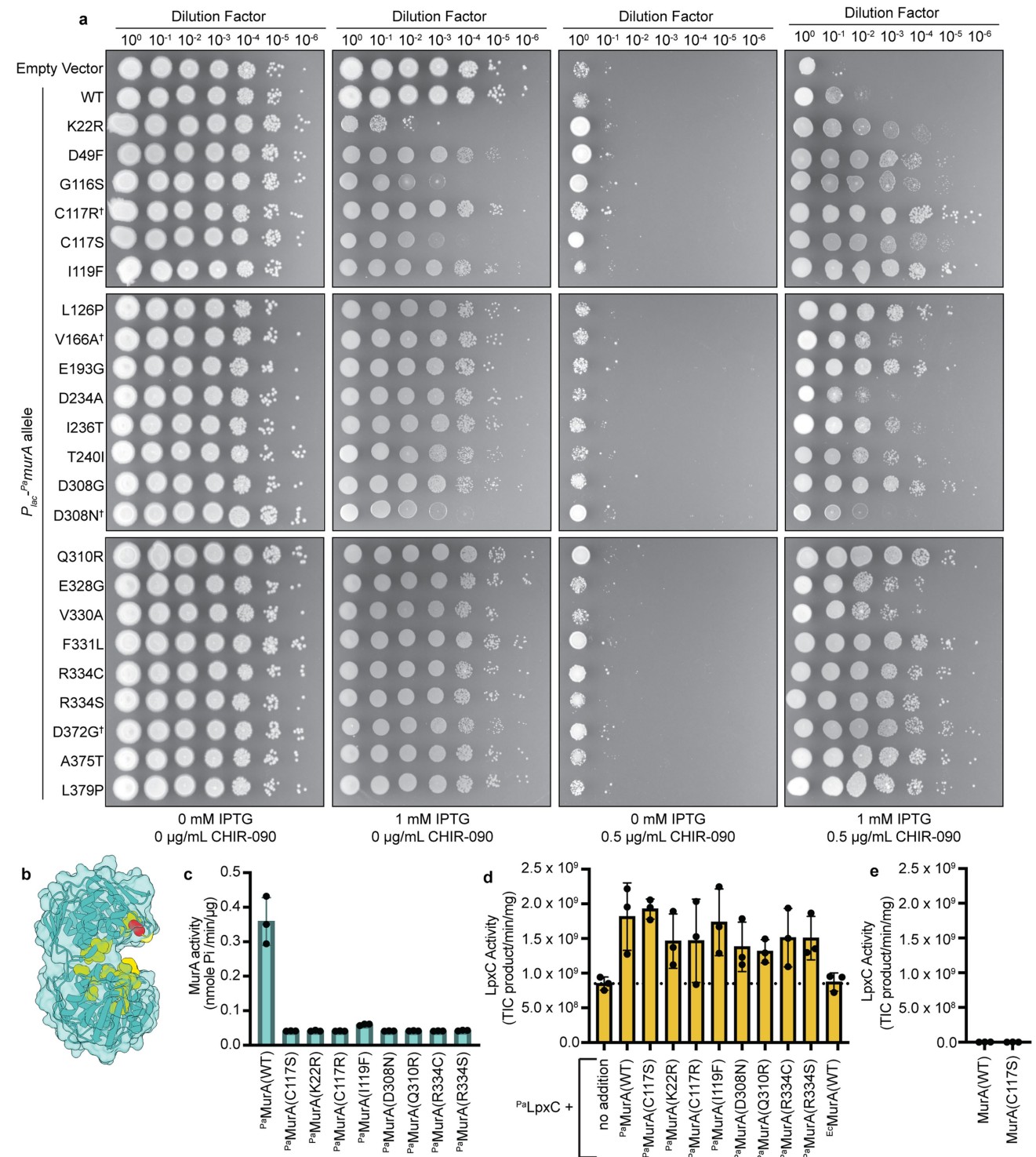

**Extended Data Fig. 6 | Mutations in the *Pa*MurA active site are toxic but can be suppressed by inhibition of *Pa*LpxC.** (a) Spot titer assay in which serial dilutions of PAO1 harboring an empty vector, one encoding *Pa*MurA(WT) or the indicated *Pa*MurA variant were plated on LB agar supplemented with IPTG and/ or CHIR-090 as indicated. Plates were incubated at 37 °C for 20 h before imaging. † indicates the presence of a silent mutation in the construct. See Table S2 for details. Data are representative of 3 biological replicates. (b) Crystal structure of *E. cloacae* MurA (**PDB** 1EJC)[14] in which residues corresponding to the identified *Pa*MurA dominant negative alleles are depicted in gold spheres, or in the case of Cys117, a red sphere. Note that the substitutions all cluster around the active site. (c) MurA activity assay in which purified *Pa*MurA variants (100 nM) were mixed with UDP-GlcNAc (1 mM) and PEP (0.5 mM) and the release of $P_i$ was measured by Lanzetta assay. Dots indicate the values obtained for three individual replicates, bars indicate the mean, and error bars represent their standard deviation. (d) Catalytic activity of purified *Pa*LpxC (100 nM) alone or in the presence of MurA variants (100 nM) assayed by conversion of UDP-3-O-(R-3-hydroxydecanoyl)-N-acetylglucosamine (*Pa*LpxC substrate) to UDP-3-O-(R-3-hydroxydecanoyl)-glucosamine (*Pa*LpxC product). Dots indicate the values obtained for three individual replicates, bars indicate the mean, and error bars represent their standard deviation. (e) LpxC enzymatic activity detected from preparation of MurA variants (100 nM) alone assayed as in panel d. Dots indicate the values obtained for three individual replicates, bars indicate the mean, and error bars represent their standard deviation.

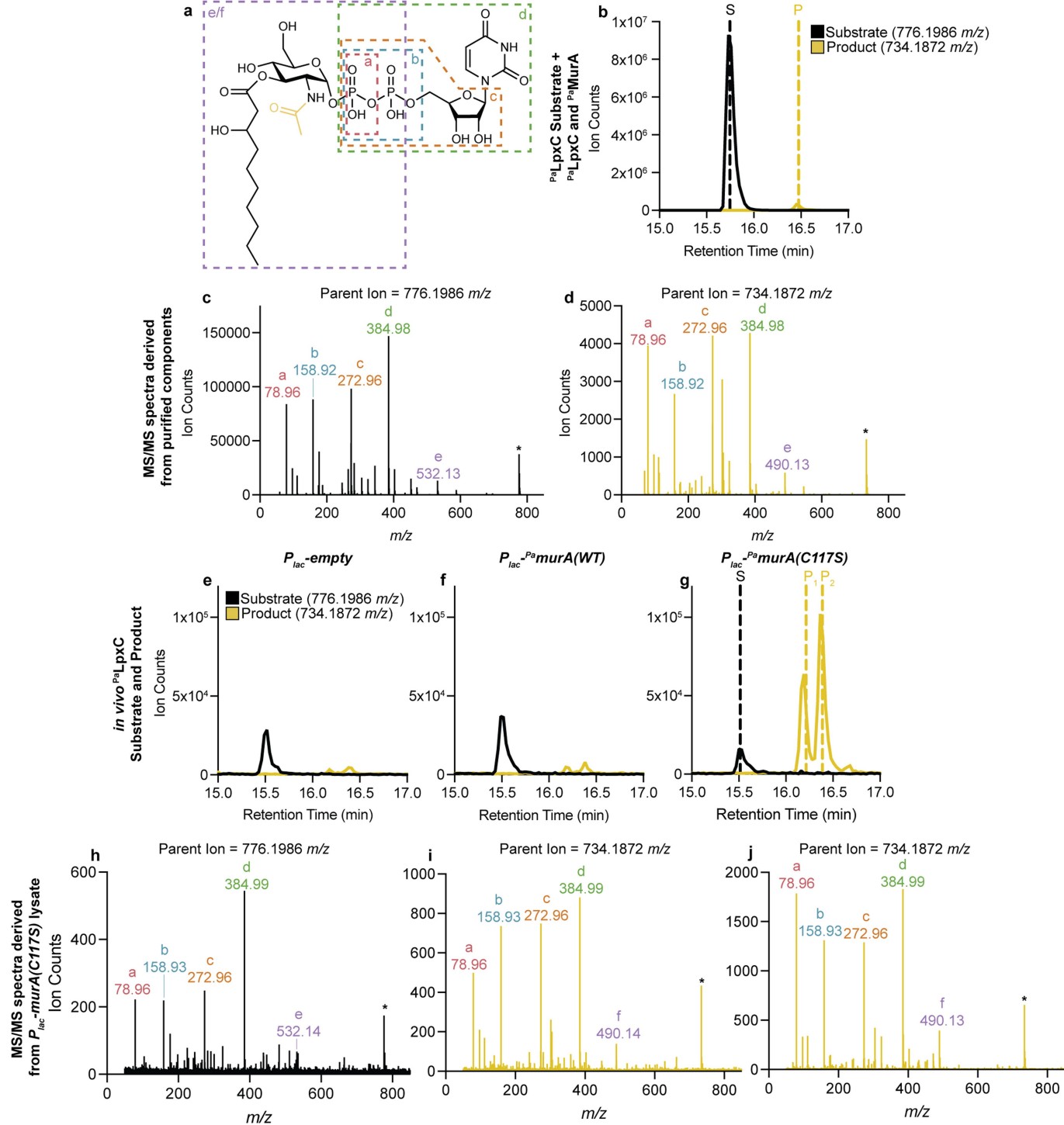

**Extended Data Fig. 7 | LC-MS/MS analysis of PaLpxC substrate and product.**
(a) Chemical structure of the PaLpxC substrate. The acetyl moiety removed by LpxC is highlighted in gold. Boxed regions indicate putative fragment ions highlighted in panels c, d, and h-j. (b) Extracted ion chromatogram (EIC) of UDP-3-O-(R-3-hydroxydecanoyl)-N-acetylglucosamine (PaLpxC substrate, 776.1986 m/z, black line) and UDP-3-O-(R-3-hydroxydecanoyl)-glucosamine (PaLpxC product, 734.1872 m/z, gold line) derived from in vitro reaction mixtures containing the PaLpxC substrate along with purified PaLpxC and PaMurA resolved using LC-MS operating in negative mode. The dashed lines indicate the peaks assigned to the PaLpxC substrate (S) and PaLpxC product (P). (c−d) MS/MS spectrum associated with the panel b peaks S and P, respectively.

The parent ion is indicated by an asterisk and fragment ions corresponding to those highlighted in panel a are labeled. (e-g) EICs PaLpxC product and PaLpxC substrate detected by LC-MS in the aqueous fraction of methanol-chloroform extracted whole cell lysates. The dashed lines indicate the peak assigned to the PaLpxC substrate (S) and PaLpxC product peaks (P₁ and P₂). Note that the PaLpxC product has been previously reported to resolve as two peaks[43], both of which were integrated to infer the relative product abundance. Data are representative of biological triplicates. (h-j) MS/MS spectrum associated with the panel g peaks S, P₁, and P₂, respectively. The parent ion is indicated by an asterisk and fragment ions corresponding to those highlighted in panel a are labeled.

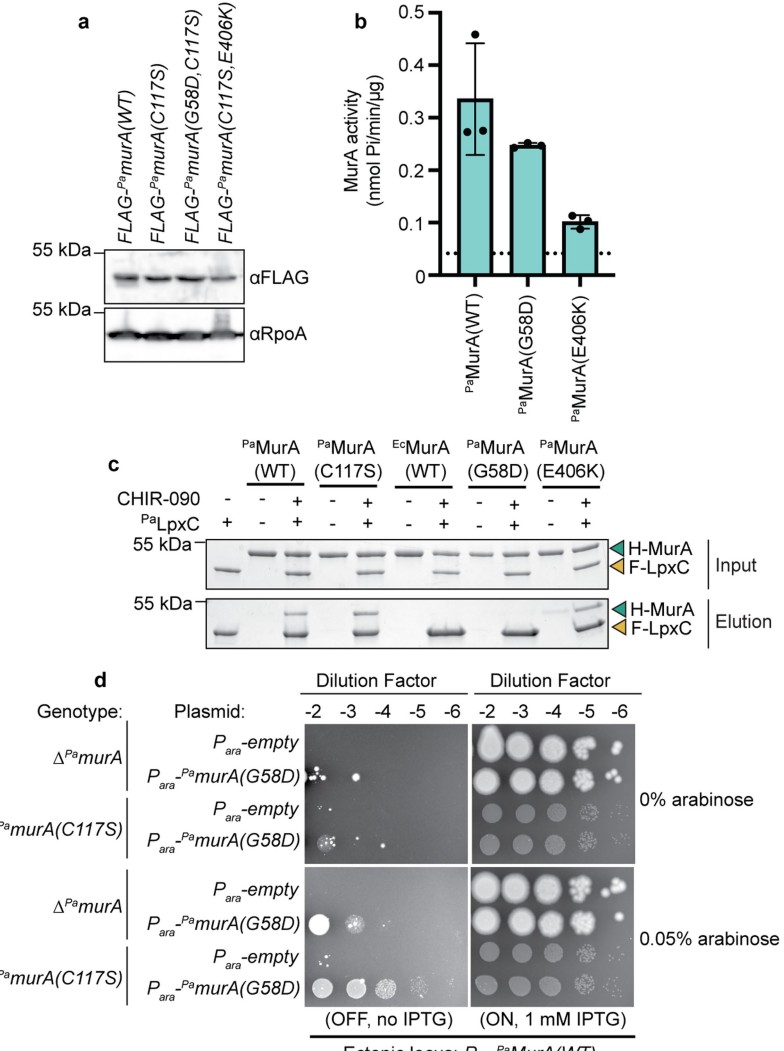

**Extended Data Fig. 8 | MurA(G58D) and MurA(E406K) impact binding and activation of ᴾᵃLpxC.** (a) Anti-FLAG immunoblot detecting F-ᴾᵃMurA variants after 1 h of induction with 1 mM IPTG. A corresponding blot for RpoA was used as a loading control. Data are representative of 3 biological replicates. (b) MurA activity assay in which purified ᴾᵃMurA variants (100 nM) were mixed with UDP-GlcNAc (1 mM) and PEP (0.5 mM) and the release of $P_i$ was measured by Lanzetta assay. Dots indicate the values obtained for three individual replicates, bars indicate the mean, and error bars represent their standard deviation. The dashed line indicates the average catalytic activity of ᴾᵃMurA(C117S) observed in Fig S7C. (c) *in vitro* pulldowns in which purified F-ᴾᵃLpxC and H-MurA variants were mixed in a 1:1 ratio in the presence or absence of CHIR-090 and processed as in Extended Data Fig. 4b. Data are representative of at least two replicates. (d) Spot titer assay in which serial dilutions of a PAO1 strain harboring a ᴾᵃ*murA* deletion or ᴾᵃ*murA(C117S)* allele at the native locus complemented by a chromosomally-integrated, IPTG-inducible copy of ᴾᵃ*murA(WT)* were plated on LB agar with the indicated supplements. As indicated, the strains also contained an empty plasmid or one encoding ᴾᵃ*murA(G58D)* under arabinose-inducible control. Plates were incubated at 37°C for 20 h before being photographed. Data are representative of 3 biological replicates. For gel source data, see Supplementary Fig. 1.

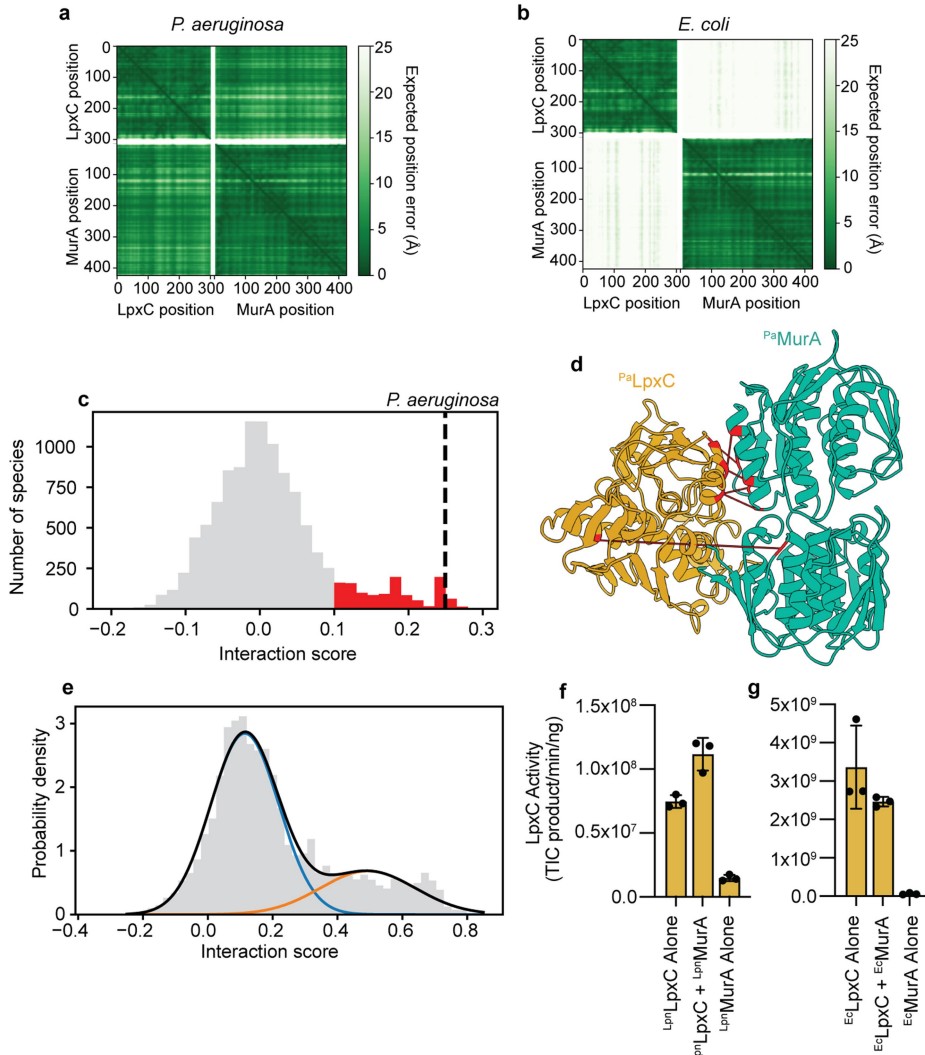

**Extended Data Fig. 9 | PaLpxC and PaMurA are predicted to interact.** (a,b) AlphaFold2 predicted aligned error matrices of the PaLpxC/PaMurA complex and EcLpxC/EcMurA complex structures, respectively. (c) Distribution of initial interaction scores among all LpxC and MurA pairs analyzed. Red bars indicate the sequences used to train the final EVcomplex model and the score corresponding to the PaLpxC/PaMurA pair is indicated by a dashed line. (d) Model structure of the PaLpxC/PaMurA complex predicted by AlphaFold2[19]. PaLpxC is represented in gold and PaMurA is represented in cyan. The top six intermolecular couplings between LpxC/MurA residues from the final EVcomplex model are highlighted in red with red lines connecting the coupling pairs. (e) Distribution of final interaction scores among all LpxC and MurA pairs analyzed. The data fit a two-component Gaussian mixture model (black line) indicating the presence of a population with low interaction scores (blue line) and high interaction scores (orange line). (f) Catalytic activity of purified LpnLpxC (100 nM) alone or in the presence of LpnMurA (200 nM) assayed by conversion of UDP-3-O-(R-3-hydroxydecanoyl)-N-acetylglucosamine to UDP-3-O-(R-3-hydroxydecanoyl)-glucosamine. Dots indicate the values obtained for three individual replicates, bars indicate the mean, and error bars represent their standard deviation. (g) Catalytic activity of purified EcLpxC (100 nM) alone or in the presence of EcMurA (200 nM) assayed by conversion of UDP-3-O-(R-3-hydroxydecanoyl)-N-acetylglucosamine to UDP-3-O-(R-3-hydroxydecanoyl)-glucosamine. Dots indicate the values obtained for three individual replicates, bars indicate the mean, and error bars represent their standard deviation.

# Reporting Summary

## Statistics

For all statistical analyses, confirm that the following items are present in the figure legend, table legend, main text, or Methods section.

| n/a | Confirmed | |
|---|---|---|
| ☐ | ☒ | The exact sample size (*n*) for each experimental group/condition, given as a discrete number and unit of measurement |
| ☐ | ☒ | A statement on whether measurements were taken from distinct samples or whether the same sample was measured repeatedly |
| ☐ | ☒ | The statistical test(s) used AND whether they are one- or two-sided *Only common tests should be described solely by name; describe more complex techniques in the Methods section.* |
| ☒ | ☐ | A description of all covariates tested |
| ☐ | ☒ | A description of any assumptions or corrections, such as tests of normality and adjustment for multiple comparisons |
| ☐ | ☒ | A full description of the statistical parameters including central tendency (e.g. means) or other basic estimates (e.g. regression coefficient) AND variation (e.g. standard deviation) or associated estimates of uncertainty (e.g. confidence intervals) |
| ☐ | ☒ | For null hypothesis testing, the test statistic (e.g. *F*, *t*, *r*) with confidence intervals, effect sizes, degrees of freedom and *P* value noted *Give P values as exact values whenever suitable.* |
| ☒ | ☐ | For Bayesian analysis, information on the choice of priors and Markov chain Monte Carlo settings |
| ☒ | ☐ | For hierarchical and complex designs, identification of the appropriate level for tests and full reporting of outcomes |
| ☒ | ☐ | Estimates of effect sizes (e.g. Cohen's *d*, Pearson's *r*), indicating how they were calculated |

*Our web collection on statistics for biologists contains articles on many of the points above.*

## Software and code

Policy information about availability of computer code

| Data collection | Covariation analysis was performed using open-source softwares Jackhmmer v3.1b2, EVcouplings v0.0.5, Python v3.8.8, FastTree v2.1.11 and AlphaFold2. |
|---|---|
| Data analysis | Statistical analyses were performed by Prism 9 (GraphPad Software, LLC.) LC/MS data were analyzed using Agilent MassHunter Workstation Qualitative Analysis software version B.06.00. Protein sequences from pulldown experiments were determined using Sequest ver 28 (rev 13). Structural data were visualized using ChimeraX version 1.1.1 and phylogenetic data were visualized using the interactive tree of life version 5. Microscopy images were processed using ImageJ2 version 2.3.0/1.53q and the MicrobeJ version 5.13I plugin. The distribution LpxC/MurA interaction scores was analyzed using sklearn v1.0.2 and ETE3 v1.3.2. |

For manuscripts utilizing custom algorithms or software that are central to the research but not yet described in published literature, software must be made available to editors and reviewers. We strongly encourage code deposition in a community repository (e.g. GitHub). See the Nature Portfolio guidelines for submitting code & software for further information.

## Data

Policy information about <u>availability of data</u>

All manuscripts must include a <u>data availability statement</u>. This statement should provide the following information, where applicable:
- Accession codes, unique identifiers, or web links for publicly available datasets
- A description of any restrictions on data availability
- For clinical datasets or third party data, please ensure that the statement adheres to our <u>policy</u>

Data used to generate graphs presented in this work are available as source data. LC-MS/MS source data, microscopy images, and computational intermediates used to derive LpxC-MurA interaction scores can be accessed using the DOI 10.5281/zenodo.7455522. Uniprot accession codes for genes used to generate LpxC-MurA interaction scores can be found in the methods and Source Data 3. All bacterial strains and plasmids developed in this study are available upon request. All code generated in this study can be accessed https://github.com/samberry19/evcomplex-interaction-scoring or by using DOI 10.5281/zenodo.7471436.

## Human research participants

Policy information about <u>studies involving human research participants and Sex and Gender in Research.</u>

| | |
|---|---|
| Reporting on sex and gender | N/A |
| Population characteristics | N/A |
| Recruitment | N/A |
| Ethics oversight | N/A |

Note that full information on the approval of the study protocol must also be provided in the manuscript.

# Field-specific reporting

Please select the one below that is the best fit for your research. If you are not sure, read the appropriate sections before making your selection.

☒ Life sciences   ☐ Behavioural & social sciences   ☐ Ecological, evolutionary & environmental sciences

For a reference copy of the document with all sections, see nature.com/documents/nr-reporting-summary-flat.pdf

# Life sciences study design

All studies must disclose on these points even when the disclosure is negative.

| | |
|---|---|
| Sample size | No statistical methods were used to predetermine sample size. In the case of microscopy experiments, all experiments were performed using greater than 100 cells per population spanning multiple fields of view, which is standard for the field. All other experiments reported bulk measurements of population wide behaviors, involving greater than 1x10^8 cells (for in vivo assays) or greater than 1 x 10^12 protein copies (for in vitro assays) as is standard for the field and each experiment was performed at least twice. |
| Data exclusions | No data were excluded from this study |
| Replication | All data are from a minimum of two independent experiments. All replication attempts were successful. |
| Randomization | Samples were not randomized in this study. Covariates were controlled by reproducing the experiments on separate days. |
| Blinding | We did not blind samples as, after each experimental setup, all measurements and analyses were performed identically across all conditions. |

# Reporting for specific materials, systems and methods

We require information from authors about some types of materials, experimental systems and methods used in many studies. Here, indicate whether each material, system or method listed is relevant to your study. If you are not sure if a list item applies to your research, read the appropriate section before selecting a response.

## Materials & experimental systems

| n/a | Involved in the study |
|-----|----------------------|
| ☐ | ☒ Antibodies |
| ☒ | ☐ Eukaryotic cell lines |
| ☒ | ☐ Palaeontology and archaeology |
| ☒ | ☐ Animals and other organisms |
| ☒ | ☐ Clinical data |
| ☒ | ☐ Dual use research of concern |

## Methods

| n/a | Involved in the study |
|-----|----------------------|
| ☒ | ☐ ChIP-seq |
| ☒ | ☐ Flow cytometry |
| ☒ | ☐ MRI-based neuroimaging |

## Antibodies

| Antibodies used | Anti-His (GenScript A00186-100); Anti-FLAG (Sigma F7425); Anti-RpoA (BioLegend 663104); Goat-anti-Mouse (Rockland 610-1302); Rabbit TrueBlot®: Anti-Rabbit IgG HRP (Rockland 18-8816-33) |
|---|---|
| Validation | Anti-His and Anti-FLAG antibodies have been used numerous times to detect His and FLAG-tagged proteins by western blot and validation information can be found at the manufacturers' websites (https://www.genscript.com/antibody/A00186-THE_His_Tag_Antibody_mAb_Mouse.html and https://www.sigmaaldrich.com/US/en/product/sigma/f7425, respectively). In addition, we verified that neither antibody reacts with Pseudomonas aeruginosa lysates derived from strains that do not encode a His- or FLAG-tagged protein (see Extended Data 4A). Monoclonal Anti-RpoA Clone 4RA2 has been used extensively in the field to detect the RNA polymerase subunit alpha in diverse gram-negative bacteria and validation can be found at the manufacturer's website (https://www.biolegend.com/it-it/products/purified-anti-e-coli-rna-polymerase-alpha-antibody-14680). Further, this antibody has been shown to react with specifically with purified P. aeruginosa PAO1 RNA polymerase (Ceyssens, P. J. et al. The Phage-Encoded N-Acetyltransferase Rac Mediates Inactivation of Pseudomonas aeruginosa Transcription by Cleavage of the RNA Polymerase Alpha Subunit. Viruses 12, doi:10.3390/v12090976 (2020).) |

