## [Peer Review File · Nature]

Manuscript Title: Coordination of bacterial cell wall and outer membrane biosynthesis

Reviewer Comments & Author Rebuttals

Reviewer Reports on the Initial Version:

Referee #1 (Remarks to the Author):

The work of Hummels et al., on the coordinated regulation of lipopolysaccharide (LPS) and peptidoglycan (PG) metabolism in *Pseudomonas aeruginosa*, demonstrates that the committed enzymes for biosynthesis of LPS (LpxC) and PG (MurA) interact both in vivo and in vitro. A key PG metabolite (UDP-MurNAc) is a known feedback inhibitor of MurA, and the authors propose that the inhibited UDP-MurNAc-MurA complex can physically interact with LpxC in a manner that stimulates LPS biosynthesis. This elegant mechanism helps to explain how LPS and PG synthesis can be balanced in order to maintain the cell envelope permeability barrier and provides insight relevant to the antibiotic treatment of bacterial infectious diseases. The activation of LpxC by an inhibited form of MurA in *Pseudomonas* appears to contrast with *Escherichia coli*, where LpxC activity is regulated instead by proteolysis in response to the accumulation of LPS in the bacterial inner membrane (such proteolytic control of LpxC does not occur in *Pseudomonas*). The described regulation of *Pseudomonas* LpxC by MurA is implied to operate in lieu of LpxC proteolysis, but in my opinion, this needs to be further clarified.

The authors profitably use *E. coli* MurA (EcMurA) as a negative control to show that it does not interact with or stimulate *Pseudomonas* LpxC (PaLpxC), as is the case with *Pseudomonas* MurA (PaMurA). What appears to be missing is the control showing what happens when *E. coli* LpxC (EcLpxC) is combined with EcMurA. Given the structural similarity between EcMurA/PaMurA and EcLpxC/PaLpxC, I am left wondering if the findings are common to both *E. coli* and *Pseudomonas aeruginosa*. What happens when EcMurA is mixed with EcLpxC? If nothing, then we can know that the findings are unique to *Pseudomonas*. If the results are recapitulated, then we can know that this regulation is more general. Either way, the findings are novel and I am hard-pressed to think of any other examples where an enzyme-inhibitor complex functions to physically modulate the activity of another regulatory enzyme that governs a parallel metabolic pathway. The authors rightly conclude that their findings “raise the possibility that there are many more regulatory interactions between enzymes involved in the biogenesis of different cell envelope components waiting to be discovered.”

Aside from my one major criticism that the interaction between EcMurA and EcLpxC should be included in this study, I offer additional minor constructive criticisms for the authors to consider:

1) Please be explicit about where His and FLAG tags are located in your constructs when you introduce them. I had to consult the primer table and align the primers with the genome sequences to convince myself that they are indeed N-terminal tags. Your nomenclature (H-PaLpxC as opposed to PaLpxC-H) hints at this point, but can by no means be taken for granted. This information is critical because the C-terminus of EcLpxC controls its degradation by FtsH. As such, a C-terminal tag would be expected to create an artefact by interfering with normal regulation.

- 2) Line 77, unable “to” associate.
- 3) Line 123, “confirmational” should read “conformational”.
- 4) Legend to Figure 2, “arounf” should be “around”.
- 5) As stated above, I encourage the authors to purify EclpxC and use it to replace PaLpxC in the experiment shown in figure 4.
- 6) In the legend to Fig S6, a footnote is needed to explain what the cross symbols represent.
- 7) In Fig S7, the Substrate peak in panel A migrates with a retention time of 16.5 minutes; in panel B 16.3 minutes; in panel C 16.1 minutes. In fact, the left shoulder of the Product peak in panel C migrates at 16.5 minutes, exactly like the Substrate peak in panel A. Clearly, there is some kind of drift going on, which needs an explanation.
- 8) In the footnotes to Table S2 you state “See Table S2 for details”. I think you mean to say “See Table S3 for details.” This same statement is also need in Figure S6 (point 6 above).
- 9) The SDS-PAGE gels showing the homogeneity of the purified His-MurA and FLAG-LpxC would be welcome additions to the supplemental material (or maybe extended data).
- 10) On page 41, “reverse-phase HPLC” should read “reversed-phase HPLC”.
- 11) On page 42, Protein was transferred “to” PVDF ...

Referee #2 (Remarks to the Author):

The cell envelope of Gram-negative bacteria contains a phospholipid inner membrane, a peptidoglycan cell wall, and an asymmetric outer membrane consisting of phospholipids in the inner leaflet and lipopolysaccharides (LPS) in the outer leaflet. The biosynthetic pathways of these molecules are well known, but how cells coordinate the biogenesis of these molecules remains inadequately understood. In Enterobacteria, control of the LPS biosynthesis is achieved through degradation of LpxC by the membrane protease FtsH. But in other bacteria, such as *Pseudomonas aeruginosa*, LpxC is not regulated by FtsH. In this manuscript, Hummels and coworkers report a surprising discovery that *Pseudomonas* LpxC (PaLpxC) forms a specific complex with *Pseudomonas* MurA (PaMurA). Through comprehensive genetic and biochemical studies, the authors establish a regulatory mechanism in that excess peptidoglycan synthesis leads to accumulation of UDP-MurNAc-bound MurA in its closed, inactive conformation, which in turn binds LpxC to activate its catalytic activity and accelerate LPS biosynthesis to achieve a balanced production of LPS and peptidoglycan. This outstanding discovery provides the first conceptual model and experimental evidence to connect LPS biosynthesis with peptidoglycan biosynthesis and is significant.

There are, however, major technical and conceptual issues that need to be addressed to support the proposed model of co-regulation. First, the interaction between wild-type PaMurA and PaLpxC appears to be very weak (Figure 1). Given that both PaMurA and PaLpxC have been purified, the authors need to use more quantitative methods (e.g., ITC) to determine the K_d value between PaMurA and PaLpxC. This information will be critical for understanding the physiological relevance of the regulation of PaLpxC by PaMurA (more below).

Second, the authors need to use the right unit to compare and report the specific activity of PaLpxC. The specific enzyme activity (or specific activity) is reported as the amount of product/time/amount of enzyme. An example is the specific activity of MurA reported in Figure 3A (nmol Pi/min/mg (murA)). Yet, in Figure 4, the activity of PaLpxC is reported as nmol product/sec, which misses the amount of enzyme used in the assay. A closer look of the assay conditions reveals that 100 μ M PaLpxC without or with 100 μ M PaMurA were used in the in vitro LpxC activity assay. 100 μ M PaLpxC is roughly 3.3 mg/mL, which is on the same scale of protein concentrations used for protein crystallization! Under steady-state or multiple-turnover conditions, the substrate concentration should be orders of magnitude higher than the enzyme concentration. What concentrations of the substrate were used in these enzymatic assays? Please note that the LpxC concentration used here (100 μ M) is $\sim 10^5$ -fold higher than a normal LpxC enzymatic assay, which typically uses sub-nM LpxC (e.g., *Biochemistry* 2005, 44, 16574-16583; *Biochemistry* 2007, 46, 3793-3802). Did authors use such a high concentration because the weak affinity between PaMurA and PaLpxC dictates that the PaMurA-activation of PaLpxC is only detectable at such a high concentration? Would such a high concentration be physiologically relevant in cells? These concerns undermine the claim of the PaMurA-mediated activation of PaLpxC. (This also emphasizes the necessity of obtaining a quantitative K_d value between WT PaMurA and PaLpxC as commented on previously.)

Additionally, Figure 4 should include PaMurA(WT) and PaMurA(C117S) alone as proper controls.

Third, the authors designed an ingenious genetic assay to isolate PaMurA mutants that enhance CHIR-090 resistance (Fig. S6). It is not explained what the "+" sign means in the figure legend. Further, it is not clear why the authors chose C117S for detailed characterization, but ignored other mutants, such as C117R, I119F, Q310R, R334C/S and others. These mutants have mild toxicity when overexpressed, but strongly rescue LpxC inhibition by CHIR-090. According to the authors' model, these mutants are expected to enhance the PaMurA-PaLpxC interaction by trapping PaMurA in a closed conformation. However, it is hard to imagine that these many mutations, all of which are located in the active site, will trap MurA in the closed conformation. In particular, it is hard to imagine that C117R, which likely creates vdw clashes with the bound substrate/product, will remain in the closed conformation and enhance PaLpxC binding. The R334C and R334S PaMurA mutants are also worrisome, as R334 forms several hydrogen bonds with active site residues, its mutation may likely distort the active site, shifting MurA into an open conformation. Another good example is the K22R mutant, which has very similar phenotypes to the C117S PaMurA mutant. K22 is directly involved in the UDP-Nacetylmuramic acid recognition (PDB: 5bq2). The K22R mutant may repel the ligand, again leading to an open conformation. These concerns can be mitigated if the authors can show that these PaMurA mutants (K22R, C117R, I119F, Q310R, R334C, and R334S) indeed enhance PaLpxC binding and activity as their model would predict.

Fourth, the observation of CHIR-090 resistance requires further consideration. The authors show that CHIR-090 inhibition of PaLpxC enhances its interaction with PaMurA, but this also implies that the PaMurA interaction would enhance the PaLpxC inhibition by CHIR-090. How such an effect would lead to CHIR-090 resistance is difficult to understand. It was reported that the presence of 4 nM CHIR-090 inhibited PaLpxC activity by over 75% (Biochemistry 2007; 46, 3793- 3802), which makes it very difficult to compensate for CHIR-090 inhibition even if PaLpxC activity is activated by PaMurA by 2-3 fold. Have the authors checked the protein level of PaLpxC in these CHIR-090 resistance cells? Is it possible that the MurA-interaction alters the PaLpxC level?

Minor: Line 146 referenced Fig. 3C. This should likely be Fig. 2C, as there is no Fig. 3C.

Overall, this manuscript reports an exciting discovery that links peptidoglycan biosynthesis with LPS biosynthesis through enzymatic activation of PaLpxC by PaMurA in its closed conformation. The proposed model is supported by elegant genetic studies, but the biochemical characterizations are inadequate. Furthermore, the impact of the discovery would be significantly enhanced if the author can demonstrate the generality of the proposed model of peptidoglycan/LPS co-regulation beyond *Pseudomonas aeruginosa*.

Referee #3 (Remarks to the Author):

In "A regulatory interaction between committed enzymes links lipopolysaccharide and peptidoglycan biogenesis in *Pseudomonas aeruginosa*," Hummels et al. explore how *Pseudomonas* regulates the amount of LPS produced by the cell. A regulatory system has been discovered in *E. coli* and other enterics that uses the proteins YejM and YciM to direct FtsH proteolysis of LPS synthesis enzyme LpxC. This regulatory system has been fairly interesting as it allows the bacterium to sense the amount of LPS at certain locations in the cell and respond accordingly to modulate LpxC levels. However, the YejM/YciM/FtsH system is not used by all Gram-negative bacteria and the authors start by studying *Pseudomonas aeruginosa*, one such bacterium. They discover that instead *Pseudomonas* senses the amount of peptidoglycan precursors and then regulates LPS synthesis to make sure these processes are balanced.

Overall, the described regulation mechanism is convincing and demonstrated through genetic and biochemical experiments, whereby Pa-MurA is inhibited if a downstream product accumulates. The inactivated conformation of Pa-MurA binds to stimulate Pa-LpxC activity to balance peptidoglycan and LPS synthesis. The genetic approach must be improved by adding quantifications of cellular LPS in mutants expected to have high or low LPS synthesis as described below in the comments. Otherwise, the approach is fairly convincing.

However, as it stands, the relevance of the story to a broad audience is not clear. There is the potential that the regulation mechanism could be broadly used across Gram-negative bacteria because (i) the previously describe YciM/YejM/FtsH regulation mechanism is not broadly conserved, and (ii) all Gram-negative bacteria have MurA and LpxC enzymes. If the authors are able to add an adequate analysis of how conserved this mechanism is, this work could merit publication in *Nature*, as it could be the more conserved method of LPS regulation.

Major items:

1. Some assessment of how conserved this mechanism is would be required to make the story of suitable broad interest for publication in Nature. It is plausible that this system could be very widespread in Gram-negative bacteria. The YciM/YejM/FtsH regulation that is already known in the field is very specific to enteric bacteria and as such yciM and yejM genes are not found outside of enterics. The newly described MurA system has the potential to be found more broadly outside enterics, as all Gram-negative bacteria have MurA for peptidoglycan synthesis. The authors could consider looking at co-variance of certain regions or residues between MurA and LpxC that are found within the Pseudomonadales, but that do not exist in enterics. Or if there are domains found in Pseudomonas MurA or LpxC that are not found in enterics. If these type of elements could be identified the authors could search through other Gram-negatives for the likelihood to have MurA regulation of LpxC.

2. It seems that quantification of cellular LPS levels (SDS-PAGE, etc.) would be required for supporting arguments from FigS2, 2A, and 3B.

3. Line 67-70 and Fig S2. The logic that is used to explain differences when overexpressing Pa-LpxC and Ec-LpxC is flawed without an immunoblot or Coomassie stain to show they are expressed at similar levels. If the Ec-LpxC naturally produces much more protein then this could explain why it was toxic while the Pa-LpxC was not. If the Ec-LpxC is producing more protein it would also explain why the leaky expression of Ec-LpxC rescued the 0.1875ug/mL Chiron-090 but the leaky expression of Pa-LpxC did not.

4. Line 84- "Of the two proteins, only PaMurA is essential for growth, making it the better candidate for an activator of the essential LpxC enzyme." Change wording. While the evidence later in the paper supports that stimulation by Pa-MurA was real, at this point in the story there is no reason to assume that a stimulator of Pa-LpxC has to be essential. It could be that Pa-LpxC has a low level of activity on its own that was sufficient to support growth, but that the stimulation by Pa-MurA or PA4701 increased activity under certain conditions. This is supported by the in vitro data in Fig 4.

Minor items

1. Fig S5: cell width measurement should be added to make the argument stronger. Also, it would be helpful for the reader if the authors would indicate the blebs on the actual figure.

2. Line 146, Figure call out should be Fig 2C not 3C.

3. Line 62: I could have easily missed this in the text, but how was delta ftsH generated? Strain table indicated no reference for this strain. Citation in the text was for a delta ftsH in E. coli.

4. Figure S3: why was murB also assayed? Was the purpose to serve as a control? Just for clarity, maybe mention in the text

5. Fig 2: Curious if the authors tested an Ec-MurA C117S variant?

6. Can the authors comment on the linear range of the LpxC activity assay shown in Figure 4? What is the dynamic range of their assay? Perhaps a CHIR-90 control could establish lower limit of the assay? Is it possible LpxC activity for C117S is saturated?

7. Does Pa MurA expressed from Pa copurify with UDP-MurNAc as has been described in *E. coli*. Could this explain why there is no obvious difference in LpxC activity comparing WT MurA and the C117S variant? If the closed MurA conformation is the preferred structure for LpxC interaction, is it possible a closed Ec MurA may also interact with LpxC?

Author Rebuttals to Initial Comments:

Responses to Referee Comments for

A regulatory interaction between committed enzymes links lipopolysaccharide and peptidoglycan biogenesis in *Pseudomonas aeruginosa*

Authors: Katherine R. Hummels, Samuel Berry, Zhaoqi Li, Atsushi Taguchi, Joseph Min, Suzanne Walker, Debora Marks, and Thomas G. Bernhardt

Referee #1 (Remarks to the Author):

The work of Hummels et al., on the coordinated regulation of lipopolysaccharide (LPS) and peptidoglycan (PG) metabolism in *Pseudomonas aeruginosa*, demonstrates that the committed enzymes for biosynthesis of LPS (LpxC) and PG (MurA) interact both in vivo and in vitro. A key PG metabolite (UDP-MurNAc) is a known feedback inhibitor of MurA, and the authors propose that the inhibited UDP-MurNAc-MurA complex can physically interact with LpxC in a manner that stimulates LPS biosynthesis. This elegant mechanism helps to explain how LPS and PG synthesis can be balanced in order to maintain the cell envelope permeability barrier and provides insight relevant to the antibiotic treatment of bacterial infectious diseases. The activation of LpxC by an inhibited form of MurA in *Pseudomonas* appears to contrast with *Escherichia coli*, where LpxC activity is regulated instead by proteolysis in response to the accumulation of LPS in the bacterial inner membrane (such proteolytic control of LpxC does not occur in *Pseudomonas*). The described regulation of *Pseudomonas* LpxC by MurA is implied to operate in lieu of LpxC proteolysis, but in my opinion, this needs to be further clarified.

The authors profitably use *E. coli* MurA (EcMurA) as a negative control to show that it does not interact with or stimulate *Pseudomonas* LpxC (PaLpxC), as is the case with *Pseudomonas* MurA (PaMurA). What appears to be missing is the control showing what happens when *E. coli* LpxC (EcLpxC) is combined with EcMurA. Given the structural similarity between EcMurA/PaMurA and EcLpxC/PaLpxC, I am left wondering if the findings are common to both *E. coli* and *Pseudomonas aeruginosa*. What happens when EcMurA is mixed with EcLpxC? If nothing, then we can know that the findings are unique to *Pseudomonas*. If the results are recapitulated, then we can know that this regulation is more general. Either way, the findings are novel and I am hard-pressed to think of any other examples where an enzyme-inhibitor complex functions to physically modulate the activity of another regulatory enzyme that governs a parallel metabolic pathway. The authors rightly conclude that their findings “raise the possibility that there are many more regulatory interactions between enzymes involved in the biogenesis of different cell envelope components waiting to be discovered.”

Aside from my one major criticism that the interaction between EcMurA and EcLpxC should be included in this study, I offer additional minor constructive criticisms for the authors to consider:

Response: We thank the reviewer for their enthusiasm for this study. In the revised manuscript, we show that ^{Ec}MurA is incapable of interacting with ^{Ec}LpxC. This result is part of a wider examination of the conservation of the regulatory mechanism we identified in *P. aeruginosa*. We used a combination of genetics and structural predictions to model the interface between ^{Pa}MurA and ^{Pa}LpxC. This model was then used to inform a bioinformatic search for other interacting enzyme pairs in other gram-negative species. Interactions were predicted in a range of different gamma- and alpha-proteobacteria. Several examples of protein pairs with a predicted interaction were shown to interact biochemically. Proteins from *E. coli* and *A. baumannii* were predicted not to interact, and this was also confirmed biochemically.

1) Please be explicit about where His and FLAG tags are located in your constructs when you introduce them. I had to consult the primer table and align the primers with the genome sequences to convince myself that they are indeed N-terminal tags. Your nomenclature (H-PaLpxC as opposed to PaLpxC-H) hints at this point, but can by no means be taken for granted. This information is critical because the C-terminus of EcLpxC controls its degradation by FtsH. As such, a C-terminal tag would be expected to create an artefact by interfering with normal regulation.

Response: All of the fusions were N-terminal, and this is now explicitly stated in the text.

2) Line 77, unable “to” associate.

Response: This section has been revised.

3) Line 123, “confirmational” should read “conformational”.

Response: This section has been revised.

4) Legend to Figure 2, “arounf” should be “around”.

Response: This section has been revised.

5) As stated above, I encourage the authors to purify EcLpxC and use it to replace PaLpxC in the experiment shown in figure 4.

Response: Please see initial response above.

6) In the legend to Fig S6, a footnote is needed to explain what the cross symbols represent.

Response: This section has been revised.

7) In Fig S7, the Substrate peak in panel A migrates with a retention time of 16.5 minutes; in panel B 16.3 minutes; in panel C 16.1 minutes. In fact, the left shoulder of the Product peak in panel C migrates at 16.5 minutes, exactly like the Substrate peak in panel A. Clearly, there is some kind of drift going on, which needs an explanation.

Response: The drift in retention time for the peaks is likely due to the crude extracts that were analyzed on the instrument, which would be expected to cause minor changes in column performance over the course of multiple runs. This information is now included in the figure legend.

8) In the footnotes to Table S2 you state “See Table S2 for details”. I think you mean to say “See Table S3 for details.” This same statement is also need in Figure S6 (point 6 above).

Response: This section has been revised.

9) The SDS-PAGE gels showing the homogeneity of the purified His-MurA and FLAG-LpxC would be welcome additions to the supplemental material (or maybe extended data).

Response: We have added an SDS-PAGE gel of all proteins used in this study (Fig S10A)

10) On page 41, “reverse-phase HPLC” should read “reversed-phase HPLC”.

Response: This section has been revised.

11) On page 42, Protein was transferred “to” PVDF ...

Response: This section has been revised.

Referee #2 (Remarks to the Author):

The cell envelope of Gram-negative bacteria contains a phospholipid inner membrane, a peptidoglycan cell wall, and an asymmetric outer membrane consisting of phospholipids in the inner leaflet and lipopolysaccharides (LPS) in the outer leaflet. The biosynthetic pathways of these molecules are well known, but how cells coordinate the biogenesis of these molecules remains inadequately understood. In Enterobacteria, control of the LPS biosynthesis is achieved through degradation of LpxC by the membrane protease FtsH. But in other bacteria, such as *Pseudomonas aeruginosa*, LpxC is not regulated by FtsH. In this manuscript, Hummels and co-workers report a surprising discovery that *Pseudomonas* LpxC (^{Pa}LpxC) forms a specific complex with *Pseudomonas* MurA (^{Pa}MurA). Through comprehensive genetic and biochemical studies, the authors establish a regulatory mechanism in that excess peptidoglycan synthesis leads to accumulation of UDP-MurNAc-bound MurA in its closed, inactive conformation, which in turn binds LpxC to activate its catalytic activity and accelerate LPS biosynthesis to achieve a balanced production of LPS and peptidoglycan. This outstanding discovery provides the first conceptual model and experimental evidence to connect LPS biosynthesis with peptidoglycan biosynthesis and is significant.

Response: We thank the reviewer for their enthusiasm for our findings. We have added a series of new results and clarified items in the text to address the issues raised by the reviewer.

There are, however, major technical and conceptual issues that need to be addressed to support the proposed model of co-regulation.

First, the interaction between wild-type ^{Pa}MurA and ^{Pa}LpxC appears to be very weak (Figure 1). Given that both ^{Pa}MurA and ^{Pa}LpxC have been purified, the authors need to use more quantitative methods (e.g., ITC) to determine the *K_d* value between ^{Pa}MurA and ^{Pa}LpxC. This information will be critical for understanding the physiological relevance of the regulation of ^{Pa}LpxC by ^{Pa}MurA (more below).

Response: We have added size exclusion chromatography analysis to show that ^{Pa}MurA and ^{Pa}LpxC indeed form a stable complex. Additionally, we have identified a ^{Pa}MurA variant that cannot interact with or activate ^{Pa}LpxC and showed that this version of ^{Pa}MurA is unable to complement a *murA* deletion in cells despite its retention of MurA catalytic activity. These added results provide strong support for the physiological relevance of the MurA-LpxC regulatory interaction.

Second, the authors need to use the right unit to compare and report the specific activity of P_a LpxC. The specific enzyme activity (or specific activity) is reported as the amount of product/time/amount of enzyme. An example is the specific activity of MurA reported in Figure 3A (nmol Pi/min/mg (murA)). Yet, in Figure 4, the activity of P_a LpxC is reported as nmol product/sec, which misses the amount of enzyme used in the assay. A closer look of the assay conditions reveals that 100 μ M P_a LpxC without or with 100 μ M P_a MurA were used in the *in vitro* LpxC activity assay. 100 μ M P_a LpxC is roughly 3.3 mg/mL, which is on the same scale of protein concentrations used for protein crystallization! Under steady-state or multiple-turnover conditions, the substrate concentration should be orders of magnitude higher than the enzyme concentration. What concentrations of the substrate were used in these enzymatic assays?

Please note that the LpxC concentration used here (100 μ M) is $\sim 10^5$ -fold higher than a normal LpxC enzymatic assay, which typically uses sub-nM LpxC (e.g., *Biochemistry* 2005, 44, 16574-16583; *Biochemistry* 2007, 46, 3793-3802). Did authors use such a high concentration because the weak affinity between P_a MurA and P_a LpxC dictates that the P_a MurA-activation of P_a LpxC is only detectable at such a high concentration? Would such a high concentration be physiologically relevant in cells? These concerns undermine the claim of the P_a MurA-mediated activation of P_a LpxC. (This also emphasizes the necessity of obtaining a quantitative K_d value between WT P_a MurA and P_a LpxC as commented on previously.)

Response: We thank the reviewer for catching our error in noting the concentrations of protein used. The correct concentrations were 100 nM of each protein in the assay with a substrate concentration of 100 μ M. Although the protein concentration used was slightly higher than the concentrations used in past studies, we think they are reasonable considering that we were not just measuring enzyme activity but also the effect of a protein-protein interaction on activity.

Additionally, Figure 4 should include P_a MurA(WT) and P_a MurA(C117S) alone as proper controls.

Response: We have added these controls.

Third, the authors designed an ingenious genetic assay to isolate P_a MurA mutants that enhance CHIR-090 resistance (Fig. S6). It is not explained what the “†” sign means in the figure legend. Further, it is not clear why the authors chose C117S for detailed characterization, but ignored other mutants, such as C117R, I119F, Q310R, R334C/S and others. These mutants have mild toxicity when overexpressed, but strongly rescue LpxC inhibition by CHIR-090.

According to the authors' model, these mutants are expected to enhance the P_a MurA- P_a LpxC interaction by trapping P_a MurA in a closed conformation. However, it is hard to imagine that these many mutations, all of which are located in the active site, will trap MurA in the closed conformation. In particular, it is hard to imagine that C117R, which likely creates vdw clashes with the bound substrate/product, will remain in the closed conformation and enhance P_a LpxC binding. The R334C and R334S P_a MurA mutants are also worrisome, as R334 forms several hydrogen bonds with active site residues, its mutation may likely distort the active site, shifting MurA into an open conformation. Another good example is the K22R mutant, which has very similar phenotypes to the C117S P_a MurA mutant. K22 is directly involved in the UDP-N-acetylmuramic acid recognition (PDB: 5bq2). The K22R mutant may repel the ligand, again

leading to an open conformation. These concerns can be mitigated if the authors can show that these ^{Pa}MurA mutants (K22R, C117R, I119F, Q310R, R334C, and R334S) indeed enhance ^{Pa}LpxC binding and activity as their model would predict.

Response: We thank the reviewer for pushing us to investigate the other ^{Pa}MurA variants. We have now tested an extensive series of the toxic ^{Pa}MurA variants for their catalytic activity and their ability to activate ^{Pa}LpxC. All of the ^{Pa}MurA mutants tested are catalytically dead yet retain LpxC activation activity, even those more likely to remain in the open conformation. Thus, activation of LpxC does not appear to strictly require the closed conformation. Based on these results, we have revised our model to indicate that all forms of MurA can activate LpxC with the potential for the closed conformation to have enhanced activity.

With this new dataset in hand, we now believe that the toxicity of ^{Pa}MurA variants does not require a specific conformation of the protein but instead that any catalytically dead enzyme will be toxic. The reason these variants of ^{Pa}MurA are toxic when overproduced but ^{Pa}MurA(WT) is not is likely related to the competition for the available UDP-GlcNAc pool. When ^{Pa}MurA(WT) is overproduced, its activation of LpxC is counterbalanced by its utilization of UDP-GlcNAc for PG synthesis. The catalytic mutants on the other hand only have LpxC activating activity and therefore promote runaway LPS synthesis at the expense of PG synthesis.

Importantly, the MurA-LpxC regulatory system can still function to balance LPS and PG synthesis even without a conformation-specific activation. With LpxC activity dependent on MurA, the number of active LpxC molecules in the cell cannot exceed the number of MurA proteins thus preventing runaway LPS synthesis and depletion of UDP-GlcNAc available for PG synthesis. Conversely, feedback inhibition of MurA and the ability of the closed form to activate LpxC prevents PG synthesis from significantly outpacing LPS production.

Fourth, the observation of CHIR-090 resistance requires further consideration. The authors show that CHIR-090 inhibition of ^{Pa}LpxC enhances its interaction with ^{Pa}MurA, but this also implies that the ^{Pa}MurA interaction would enhance the ^{Pa}LpxC inhibition by CHIR-090. How such an effect would lead to CHIR-090 resistance is difficult to understand. It was reported that the presence of 4 nM CHIR-090 inhibited ^{Pa}LpxC activity by over 75% (*Biochemistry* 2007; 46, 3793- 3802), which makes it very difficult to compensate for CHIR-090 inhibition even if ^{Pa}LpxC activity is activated by ^{Pa}MurA by 2-3 fold. Have the authors checked the protein level of ^{Pa}LpxC in these CHIR-090 resistance cells? Is it possible that the MurA-interaction alters the ^{Pa}LpxC level?

Response: Figure S5 includes immunoblots of whole cell lysates of cultures expressing either MurA(WT) or MurA(C117S) grown in the absence or presence of CHIR-090 and there is no apparent difference in ^{Pa}LpxC levels in any condition. Thus, increased resistance to CHIR-090 is not mediated by elevated ^{Pa}LpxC levels.

The enhanced stability of MurA-LpxC complexes in the presence of CHIR-090 suggests that the drug-bound LpxC protein is in its enzymatically active state, which would enhance its binding to the MurA activator at equilibrium in vitro. However, in the cell, CHIR-090 is in competition with

substrate and the reactions are performed under steady-state not equilibrium conditions. Thus, one possible explanation for the CHIR-090 resistance promoted by the overproduction of MurA variants is that the activated LpxC molecules have a higher affinity for substrate than CHIR-090. We think this would be an interesting hypothesis to test in the future but feel that resolution of this question is not required to support our overall conclusions or regulatory model.

Minor: Line 146 referenced Fig. 3C. This should likely be Fig. 2C, as there is no Fig. 3C.

Response: This section has been revised.

Overall, this manuscript reports an exciting discovery that links peptidoglycan biosynthesis with LPS biosynthesis through enzymatic activation of Pa LpxC by Pa MurA in its closed conformation. The proposed model is supported by elegant genetic studies, but the biochemical characterizations are inadequate. Furthermore, the impact of the discovery would be significantly enhanced if the author can demonstrate the generality of the proposed model of peptidoglycan/LPS co-regulation beyond *Pseudomonas aeruginosa*.

Response: We hope the extensive additional biochemical and genetic studies are more than adequate to satisfy the reviewer's concerns. We have also added significant new information related to the conservation of the activation mechanism. Using a combination of genetics and structural predictions to model the interface between Pa MurA and Pa LpxC, we were able to perform a bioinformatic search for other interacting enzyme pairs in other gram-negative species. Interactions were predicted in a range of different gamma- and alpha-proteobacteria. Biochemical studies were then performed to show that several examples of protein pairs with a predicted interaction do indeed interact whereas those with a low interaction score were shown not to form a complex. Thus, the regulation of LpxC by MurA is likely to be utilized by a range of other bacteria beyond *P. aeruginosa*.

Referee #3 (Remarks to the Author):

In "A regulatory interaction between committed enzymes links lipopolysaccharide and peptidoglycan biogenesis in *Pseudomonas aeruginosa*," Hummels et al. explore how *Pseudomonas* regulates the amount of LPS produced by the cell. A regulatory system has been discovered in *E. coli* and other enterics that uses the proteins YejM and YciM to direct FtsH proteolysis of LPS synthesis enzyme LpxC. This regulatory system has been fairly interesting as it allows the bacterium to sense the amount of LPS at certain locations in the cell and respond accordingly to modulate LpxC levels. However, the YejM/YciM/FtsH system is not used by all Gram-negative bacteria and the authors start by studying *Pseudomonas aeruginosa*, one such bacterium. They discover that instead *Pseudomonas* senses the amount of peptidoglycan precursors and then regulates LPS synthesis to make sure these processes are balanced.

Overall, the described regulation mechanism is convincing and demonstrated through genetic and biochemical experiments, whereby Pa-MurA is inhibited if a downstream product accumulates. The inactivated conformation of Pa-MurA binds to stimulate Pa-LpxC activity to balance peptidoglycan and LPS synthesis. The genetic approach must be improved by adding quantifications of cellular LPS in mutants expected to have high or low LPS synthesis as described below in the comments. Otherwise, the approach is fairly convincing.

However, as it stands, the relevance of the story to a broad audience is not clear. There is the

potential that the regulation mechanism could be broadly used across Gram-negative bacteria because (i) the previously describe YciM/YejM/FtsH regulation mechanism is not broadly conserved, and (ii) all Gram-negative bacteria have MurA and LpxC enzymes. If the authors are able to add an adequate analysis of how conserved this mechanism is, this work could merit publication in Nature, as it could be the more conserved method of LPS regulation.

Response: We thank the reviewer for their enthusiasm for our study. The revised version shows that the regulatory mechanism we have identified is indeed likely to be used by a variety of gram-negative organisms.

Major items:

1. Some assessment of how conserved this mechanism is would be required to make the story of suitable broad interest for publication in Nature. It is plausible that this system could be very wide-spread in Gram-negative bacteria. The YciM/YejM/FtsH regulation that is already know in the field is very specific to enteric bacteria and as such yciM and yejM genes are not found outside of enterics. The newly described MurA system has the potential to be found more broadly outside enterics, as all Gram-negative bacteria have MurA for peptidoglycan synthesis. The authors could consider looking at co-variance of certain regions or residues between MurA and LpxC that are found within the Pseudomonadales, but that do not exist in enterics. Or if there are domains found in Pseudomonas MurA or LpxC that are not found in enterics. If these type of elements could be identified the authors could search through other Gram-negatives for the likelihood to have MurA regulation of LpxC.

Response: As the reviewer suggests, the MurA-LpxC regulatory system has the potential to be broadly conserved in gram-negative bacteria. Because MurA and LpxC are found in all gram-negative bacteria yet only a subset of them are likely to interact, standard methods of co-variation analysis were not fruitful. The interaction signal from covarying residues gets lost in the noise from instances where LpxC and MurA do not interact. To circumvent this issue, we used a genetically validated AlphaFold prediction of the ^{Pa}MurA-^{Pa}LpxC complex to define the likely protein-protein interaction interface. The predicted interface was then used to inform a bioinformatic search of LpxC/MurA pairs for covarying residues in this region from 8302 diverse gram-negative organisms. The analysis and subsequent biochemical validation revealed that the interaction is conserved in a range of gamma- and alpha-proteobacteria, indicating that the regulatory mechanism is likely to be utilized in many different gram-negative bacteria.

2. It seems that quantification of cellular LPS levels (SDS-PAGE, etc.) would be required for supporting arguments from FigS2, 2A, and 3B.

Response: We have added analysis of LPS levels by SDS-PAGE + silver stain as suggested (Fig S3). We found that ectopic expression of ^{Ec}LpxC or ^{Pa}MurA(C117S) increased levels of LPS and that depletion of ^{Pa}MurA(WT) decreased levels of LPS, as predicted by our model.

3. Line 67-70 and Fig S2. The logic that is used to explain differences when overexpressing Pa-LpxC and Ec-LpxC is flawed without an immunoblot or Coomassie stain to show they are expressed at similar levels. If the Ec-LpxC naturally produces much more protein then this could explain why it was toxic while the Pa-LpxC was not. If the Ec-LpxC is producing more protein it would also explain why the leaky expression of Ec-LpxC rescued the 0.1875ug/mL Chiron-090 but the leaky expression of Pa-LpxC did not.

Response: Western blot analysis of ^{Ec}LpxC or ^{Pa}LpxC ectopically expressed in *P. aeruginosa* showed that ^{Ec}LpxC was expressed much more poorly than ^{Pa}LpxC (see new Fig S2B) and thus the stronger phenotype associated with ^{Ec}LpxC expression cannot be attributed to it being produced at higher levels. It is notable that the observed molecular weights of ^{Ec}LpxC and ^{Pa}LpxC are different, despite the predicted molecular weights being nearly identical. This difference is not due to the heterologous expression of ^{Ec}LpxC in *P. aeruginosa*, however, as ^{Ec}LpxC and ^{Pa}LpxC purified from *E. coli* exhibit similar observed molecular weight differences (Fig S10A).

4. Line 84- “Of the two proteins, only PaMurA is essential for growth, making it the better candidate for an activator of the essential LpxC enzyme.” Change wording. While the evidence later in the paper supports that stimulation by Pa-MurA was real, at this point in the story there is no reason to assume that a stimulator of Pa-LpxC has to be essential. It could be that Pa-LpxC has a low level of activity on its own that was sufficient to support growth, but that the stimulation by Pa-MurA or PA4701 increased activity under certain conditions. This is supported by the in vitro data in Fig 4.

Response: This portion of the results has been rewritten accordingly.

Minor items

1. Fig S5: cell width measurement should be added to make the argument stronger. Also, it would be helpful for the reader if the authors would indicate the blebs on the actual figure.

Response: We have quantified cell width for the strains used in Figure S4 and added these data to the figure. The analysis supports the previous conclusions of the paper: depletion of MurA phenocopies the simultaneous chemical inhibition of LpxC and MurA. We have also added carrots to highlight cell blebs in the Fosfomycin and MurB depletion conditions.

2. Line 146, Figure call out should be Fig 2C not 3C.

Response: This section has been revised.

3. Line 62: I could have easily missed this in the text, but how was delta ftsH generated? Strain table indicated no reference for this strain. Citation in the text was for a delta ftsH in *E. coli*.

Response: The $\Delta ftsH$ allele was generated by allelic replacement using pKH3, which was described in the methods. The specific strain construction details can be found in the 6-his-^{Pa}LpxC section under allelic replacement heading in the methods.

4. Figure S3: why was murB also assayed? Was the purpose to serve as a control? Just for clarity, maybe mention in the text

Response: Assaying the effect of MurB depletion was indeed used as a control. We now mention this in the figure legend.

5. Fig 2: Curious if the authors tested an Ec-MurA C117S variant?

Response: Given that ^{Ec}MurA does not interact with ^{Ec}LpxC or ^{Pa}LpxC, we did not test this variant.

6. Can the authors comment on the linear range of the LpxC activity assay shown in Figure 4? What is the dynamic range of their assay? Perhaps a CHIR-90 control could establish lower limit of the assay? Is it possible LpxC activity for C117S is saturated?

Response: Under our conditions, the activity of LpxC is linear within the time frame of the experiments. We have added data showing this linearity (Figure S10B).

7. Does Pa MurA expressed from Pa copurify with UDP-MurNAc as has been described in E. coli. Could this explain why there is no obvious difference in LpxC activity comparing WT MurA and the C117S variant? If the closed MurA conformation is the preferred structure for LpxC interaction, is it possible a closed Ec MurA may also interact with LpxC?

Response: Upon analysis of additional ^{Pa}MurA variants and their effect on LpxC activation, we now do not think that the closed conformation of MurA is required for the activation mechanism. Please see the response to Reviewer #2's third major comment.

Reviewer Reports on the First Revision:

Referee #1 (Remarks to the Author):

The authors have provided a fair rebuttal of my prior criticisms. The revised manuscript is much improved, and is very convincing and unclouded by any ambiguity. My specific requests to clarify certain statistical anomalies have been clearly addressed. I consider this work to be a real triumph of enzymology and bacterial genetics.

I would be remiss if I neglected to mention that I consider it customary to describe R-3-hydroxydecanoyl as one word, rather than as R-3-hydroxy-decanoyl as employed consistently with a hyphen by the authors. I wonder what would the copy editors will say.

Referee #2 (Remarks to the Author):

RE: 2021-07-11441A

The revised manuscript by Hummels and colleagues clarified confusions and corrected errors in the initial submission. Inclusion of addition genetic, bioinformatics, and biochemical data (particularly Figure 5) further strengthens the implication of the LpxC regulation by MurA. However, several issues remain to be addressed before this manuscript is deemed suitable for publication in Nature.

Major:

The generality of the LpxC regulation by MurA is an important message of this manuscript. Hence, it is inadequate for the authors to use the MurA-LpxC pulldown as the surrogate to address this request by multiple reviewers. The authors need to show that a similar C117S mutant in other MurA orthologs similarly leads to a dramatic increase of the LpxC product/substrate ratio in order to establish the biological consequence of LpxC misregulation by the corresponding MurA mutant in these bacteria.

Figures 2 and 4: The PaMurA(E406K) result appears to contradict the proposed model of PaLpxC activation by PaMurA. It is not clear how this PaMurA mutant would retain its ability to interact with PaLpxC but abolish PaLpxC activation. The authors conveniently suggest a model that the E406K PaMurA mutant inhibits PaLpxC at a later conformational step without providing any tangible evidence. Is the PaMurA E406K mutation located near the LpxC active site in the AlphaFold model? Does AlphaFold predict different PaLpxC-binding modes for the PaMurA WT enzyme and the E406K and G58D mutants?

Figure S7: Please report the predicted and experimentally measured exact m/z values and additionally include fragmentation data (LC-MS/MS) of the LpxC substrate and product. Since this is a critical piece of information to establish the regulation of PaLpxC by PaMurA, it is important for the authors to include more detailed mass spec analysis to support their conclusion.

Since the authors have purified PaMurA and PaLpxC, it is unclear why the authors decline to provide quantitative binding data between PaMurA and PaLpxC. This does not have to come from ITC measurements. SPR or BLI measurements would provide similar information.

Minor issues:

Figure 2: Please check for typos in the column labeling. “PaMruA(WT)” and “PaMruA(E406k)” are likely typos. Additionally, please consider moving the PaMurA(C117S) result from Figure S6C to Figure 2, as this mutant is used in Figure 3 to establish the in vivo consequence of PaLpxC misregulation by the PaMurA C117S mutant.

Figure S2B: The abnormal shifts and double bands of His-EcLpxC expressed in *P. aeruginosa* is worrisome. Could the authors use other LpxC enzymes, such as *Acinetobacter* LpxC, as controls?

Figure S6D: There is no caption for Figure S6D. It is not clear what this panel is intended to show.

Line 193: The authors stated that “Both residues G58 and E406 lie on the same surface of the PaMurA structure distinct from the active site, suggesting that they may comprise the PaLpxC binding interface (Fig 4C).” This statement contradicts the observation that “H-PaMurA(E406K) exhibited binding activity similar to H-PaMurA(WT).”

Based on Figure 5, the authors’ statement that “a substantial group of gamma- and alphaproteobacteria control LpxC through interaction with and activation by MurA (Fig 5A)” seems an overstatement. This reviewer would suggest that the authors provide a quantitative number (e.g. ~25%) to avoid misinterpretation. Also, as illustrated by the PaMurA(E406K) result, the MurA-LpxC interaction does not necessarily lead to LpxC activation. Hence, it is important for the authors to show that these additional MurA orthologs indeed enhance the activity of the targeting LpxC enzymes.

Referee #3 (Remarks to the Author):

The authors have done a thorough revision and have addressed all previous concerns.

Author Rebuttals to First Revision:

Responses to Reviewer Comments:

Referee #1 (Remarks to the Author):

The authors have provided a fair rebuttal of my prior criticisms. The revised manuscript is much improved, and is very convincing and unclouded by any ambiguity. My specific requests to clarify certain statistical anomalies have been clearly addressed. I consider this work to be a real triumph of enzymology and bacterial genetics.

RESPONSE: We thank the reviewer for their encouraging words and enthusiasm for our work.

I would be remiss if I neglected to mention that I consider it customary to describe R-3-hydroxydecanoyl as one word, rather than as R-3-hydroxy-decanoyl as employed consistently with a hyphen by the authors. I wonder what would the copy editors will say.

Referee #2 (Remarks to the Author):

RE: 2021-07-11441A

The revised manuscript by Hummels and colleagues clarified confusions and corrected errors in the initial submission. Inclusion of addition genetic, bioinformatics, and biochemical data (particularly Figure 5) further strengthens the implication of the LpxC regulation by MurA. However, several issues remain to be addressed before this manuscript is deemed suitable for publication in Nature.

RESPONSE: We were glad to see that the reviewer thinks that the addition of new genetic, biochemical, and bioinformatic data to the revised report has clarified and strengthened our findings describing the regulation of LpxC by MurA. We thank you for the careful review of our findings.

Major:

The generality of the LpxC regulation by MurA is an important message of this manuscript. Hence, it is inadequate for the authors to use the MurA-LpxC pulldown as the surrogate to address this request by multiple reviewers. The authors need to show that a similar C117S mutant in other MurA orthologs similarly leads to a dramatic increase of the LpxC product/substrate ratio in order to establish the biological consequence of LpxC misregulation by the corresponding MurA mutant in these bacteria.

RESPONSE: We agree with the reviewer that the potential conservation of the LpxC-MurA regulatory connection is an important message of our paper. However, we respectfully disagree with the reviewer that it is necessary to reconstitute this regulation in another organism. Of course, such data would be nice to add if it were a simple experiment. But, in this case, the reviewer is setting the bar unreasonably high with such a request. It would require us to establish genetic methods in organisms that we have no experience with and that are not that genetically tractable to begin with. Additionally, showing that such regulation is operating in a few other organisms does not significantly add to the generality argument. Given the vast number of different bacteria out there, a few more species is a drop in the bucket.

We would also like to push back on the reviewer's implied argument that demonstration of a broad distribution for the LpxC-MurA regulatory system is a critical component of the impact of our report rather than a nice addition. The data in *P. aeruginosa* alone provide convincing evidence that the committed enzymes of two biosynthetic pathways are connected by a positive regulatory interaction. As far as we are aware, this is a unique and novel finding in biology for metabolic regulation. In addition, this interaction happens to be between two key drug targets in one of the most notorious and problematic drug-resistant bacteria out there. In our opinion, these two points alone provide a level of impact worthy of publication in this esteemed journal. Nevertheless, we took the comments of all three reviewers

related to the conservation issue to heart and embarked on a difficult and thorough bioinformatic analysis for the revision that was buttressed with biochemical data to make a reasonable case for the LpxC-MurA regulatory system being operative in a range of proteobacteria. We thus feel we have gone above and beyond in addressing the reviewers' concerns. The other two reviewers are in full agreement with us. We therefore hope that you will reconsider this point.

Figures 2 and 4: The PaMurA(E406K) result appears to contradict the proposed model of PaLpxC activation by PaMurA. It is not clear how this PaMurA mutant would retain its ability to interact with PaLpxC but abolish PaLpxC activation. The authors conveniently suggest a model that the E406K PaMurA mutant inhibits PaLpxC at a later conformational step without providing any tangible evidence. Is the PaMurA E406K mutation located near the LpxC active site in the AlphaFold model? Does AlphaFold predict different PaLpxC-binding modes for the PaMurA WT enzyme and the E406K and G58D mutants?

RESPONSE: We respectfully disagree with the reviewer's point here. It is not uncommon for mutants in a regulatory protein to be isolated that remain capable of interacting with a partner but fail to induce whatever conformational change is needed to mediate the regulation. We therefore think the results with the E406K mutant remain in full agreement with our model and are relatively straightforward to interpret. Determining the precise conformational changes in LpxC induced by MurA to activate LPS synthesis and why the E406K mutation fails to elicit this change are well beyond the scope of this report and best left to a dedicated structural analysis that will likely require years of additional work.

Figure S7: Please report the predicted and experimentally measured exact m/z values and additionally include fragmentation data (LC-MS/MS) of the LpxC substrate and product. Since this is a critical piece of information to establish the regulation of PaLpxC by PaMurA, it is important for the authors to include more detailed mass spec analysis to support their conclusion.

RESPONSE: We have added the theoretical and experimental m/z values for the LC-MS analysis of LpxC substrate and product to the figure as requested. We do not have fragmentation data for this experiment, but do not feel it is necessary to support our conclusion. The m/z values of the molecules we detect are very close to the theoretical values and are consistent with our purified authentic standard for the substrate. Also, the LC-MS data are consistent with our extensive genetic data and the observed increase in total cellular LPS in cells expressing MurA(C117S). It is hard to imagine how the conclusions on this point could be supported any more strongly.

Since the authors have purified PaMurA and PaLpxC, it is unclear why the authors decline to provide quantitative binding data between PaMurA and PaLpxC. This does not have to come from ITC measurements. SPR or BLI measurements would provide similar information.

RESPONSE: We tried BLI but had trouble getting reliable interaction data. We suspect that the orientation of the tagged proteins on the surface of the biosensor somehow interferes with the interaction. ITC requires a large amount of protein, and our yields of LpxC were low such that this was not feasible. In any case, the data we present, including pull-downs, SEC, structural modeling, and genetics all strongly support that the two proteins interact as part of a physiologically relevant regulatory system. We don't think putting a number to the interaction affinity adds much in the way of important information at this point, especially since the affinity would be measured in artificial conditions outside the cell. In our opinion, the genetic data make a much better case for the physiological relevance of the interaction than any K_d measurement.

Minor issues:

Figure 2: Please check for typos in the column labeling. "PaMruA(WT)" and "PaMruA(E406k)" are likely typos. Additionally, please consider moving the PaMurA(C117S) result from Figure S6C to Figure 2, as this mutant is used in Figure 3 to establish the in vivo consequence of PaLpxC misregulation by the PaMurA C117S mutant.

RESPONSE: We have corrected these typos and moved the PaMurA(C117S) data to Figure 1C as requested.

Figure S2B: The abnormal shifts and double bands of His-EcLpxC expressed in *P. aeruginosa* is worrisome. Could the authors use other LpxC enzymes, such as Acinetobacter LpxC, as controls?

RESPONSE: We agree that the abnormal MW of ^EC_{LpxC} when expressed in *P. aeruginosa* is strange, but we don't think it is worrisome. The toxicity of expressing this variant in *P. aeruginosa* can be alleviated by the LpxC inhibitor CHIR-090, indicating that it is working as expected. Also, this data was part of an early experiment that merely helped us start on the road to discover the interaction between the *P. aeruginosa* LpxC and MurA enzymes. It is therefore not critical for our ultimate conclusions.

Figure S6D: There is no caption for Figure S6D. It is not clear what this panel is intended to show.

RESPONSE: We have added a corresponding caption for Figure S6D.

Line 193: The authors stated that “Both residues G58 and E406 lie on the same surface of the PaMurA structure distinct from the active site, suggesting that they may comprise the PaLpxC binding interface (Fig 4C).” This statement contradicts the observation that “H-PaMurA(E406K) exhibited binding activity similar to H-PaMurA(WT).”

RESPONSE: See response to the related comment above. Not all changes in a binding interface will disrupt binding of protein partners. We therefore think that this result is perfectly consistent with the statement.

Based on Figure 5, the authors' statement that “a substantial group of gamma- and alphaproteobacteria control LpxC through interaction with and activation by MurA (Fig 5A)” seems an overstatement. This reviewer would suggest that the authors provide a quantitative number (e.g. ~25%) to avoid misinterpretation. Also, as illustrated by the PaMurA(E406K) result, the MurA-LpxC interaction does not necessarily lead to LpxC activation. Hence, it is important for the authors to show that these additional MurA orthologs indeed enhance the activity of the targeting LpxC enzymes.

RESPONSE: We have expanded our computational analysis of LpxC/MurA covariation to include an estimation of the number of alpha- and gamma- proteobacteria where the interaction is conserved. Based on the bimodal distribution of the LpxC/MurA interaction scores, we estimate that 48% of gammaproteobacterial and 35% of alphaproteobacteria encode LpxC/MurA pairs that are capable of interacting. We have added these values to the main text as suggested and included a graph of the distribution of final interaction scores to support this estimation in Extended Data 10E. We have also performed biochemical assays that show that LpxC from *Legionella pneumophila* (an organism with a high interaction score) is stimulated by its cognate MurA in vitro. In contrast, the activity of LpxC from *E. coli* (with a low interaction score) was unaffected by the presence of MurA.

Referee #3 (Remarks to the Author):

The authors have done a thorough revision and have addressed all previous concerns.

RESPONSE: We thank the reviewer for their careful review and we are glad that our revision addressed all of your previous concerns.

Reviewer Reports on the Second Revision:

Referee #2 (Remarks to the Author):

This reviewer can agree with the authors' statements about the technical difficulties in obtaining additional genetic data and quantitative PaLpxC-PaMurA binding data. However, this reviewer cannot agree with the authors' refusal to provide the LC-MS/MS data for the in vivo extracted LpxC product in Extended Data Figure 8i.

The authors' biochemical and genetic data are outstanding in connecting the PaMurA-PaLpxC interaction with the regulation of LpxC activity in vitro. However, the dramatic accumulation of the LpxC product in cells (Fig. 2B) is a key argument that connects in vitro observations with in vivo phenotypes.

The raw data supporting Fig. 2B are presented in the Extended Data Figure 8i. Unfortunately, such a critical result lacks rigorous analysis. In this experiment, the aqueous fraction of the methanol-chloroform extraction of the whole cell lysate was subject to the LC-MS analysis. Due to the presence of a large number of metabolites and lipid species, there is no guarantee that the extracted ion current at $m/z=734.1944$ represents the ion current of the LpxC product. Therefore, a fragmentation analysis is a litmus test to ensure that the LpxC product is indeed represented by the extracted ion current as stated. Without such validation, it is possible that some other metabolites or lipid species having a similar m/z ratio could mislead the authors. Additionally, considering the inconsistent retention time and twin peak heights of the ion current of the LpxC product in vitro (first peak higher) versus in vivo extract (second peak higher), it would be prudent for the authors to verify that their observed species is indeed the LpxC product. This reviewer feels such a result is a cornerstone of the authors' argument and needs to be built on a solid foundation. This experiment is also not hard. The authors should already have the fragmentation data, especially considering the extremely high signals reported by the authors. If the authors need to repeat this experiment, it can be done within a couple of days at the maximum.

Referee #4 (Remarks to the Author):

I agree with the comments of the reviewer that further MS analysis needs to be completed to strengthen their argument that the in vivo analysis exactly represents the in vitro analysis. Additionally, the following two statements by the authors in the response to reviewers' comments give me further pause about the rigor of the MS analysis. They are using a MS instrument with high mass accuracy and statements such as "very close" and "slightly altering column performance" can easily be address with further MS analysis. I do realize that this might be asking for more work to be carried out (which may be hard to do) but qualifiers for MS results, where structural analysis and new columns shouldn't be an impedance, should be possible.

- 1) The m/z values of the molecules we detect are very close to the theoretical values and are consistent with our purified authentic standard for the substrate
- 2) Drift in the retention times of the PaLpxC substrate and product between samples is likely a result

of the samples slightly altering column performance.

Author Rebuttals to Second Revision:

Referee #2 (Remarks to the Author):

This reviewer can agree with the authors' statements about the technical difficulties in obtaining additional genetic data and quantitative PaLpxC-PaMurA binding data. However, this reviewer cannot agree with the authors' refusal to provide the LC-MS/MS data for the *in vivo* extracted LpxC product in Extended Data Figure 8i.

The authors' biochemical and genetic data are outstanding in connecting the PaMurA-PaLpxC interaction with the regulation of LpxC activity *in vitro*. However, the dramatic accumulation of the LpxC product in cells (Fig. 2B) is a key argument that connects *in vitro* observations with *in vivo* phenotypes.

The raw data supporting Fig. 2B are presented in the Extended Data Figure 8i. Unfortunately, such a critical result lacks rigorous analysis. In this experiment, the aqueous fraction of the methanol-chloroform extraction of the whole cell lysate was subject to the LC-MS analysis. Due to the presence of a large number of metabolites and lipid species, there is no guarantee that the extracted ion current at $m/z=734.1944$ represents the ion current of the LpxC product. Therefore, a fragmentation analysis is a litmus test to ensure that the LpxC product is indeed represented by the extracted ion current as stated. Without such validation, it is possible that some other metabolites or lipid species having a similar m/z ratio could mislead the authors. Additionally, considering the inconsistent retention time and twin peak heights of the ion current of the LpxC product *in vitro* (first peak higher) versus *in vivo* extract (second peak higher), it would be prudent for the authors to verify that their observed species is indeed the LpxC product. This reviewer feels such a result is a cornerstone of the authors' argument and needs to be built on a solid foundation. This experiment is also not hard. The authors should already have the fragmentation data, especially considering the extremely high signals reported by the authors. If the authors need to repeat this experiment, it can be done within a couple of days at the maximum.

RESPONSE: We thank the reviewer for their enthusiasm for our biochemical and genetic data and for pushing us to add more rigor to the LC/MS analysis of LpxC substrate and product *in vivo*. We have repeated this analysis and obtained fragmentation data for the substrate and product as requested. We present this new data in the revised Extended Data Figure 8. There we show that the fragmentation pattern for our authentic LpxC substrate and the product derived from *in vitro* reactions with purified LpxC match the patterns for the peaks we assign as LpxC substrate and product from the cell extracts. Therefore, we can confidently assign the extracted ion chromatograms to the LpxC substrate and product. Thanks to the reviewer's request, the conclusions based on the *in vivo* analysis of LpxC activity are greatly strengthened.

Referee #4 (Remarks to the Author):

I agree with the comments of the reviewer that further MS analysis needs to be completed to strengthen their argument that the *in vivo* analysis exactly represents the

in vitro analysis. Additionally, the following two statements by the authors in the response to reviewers' comments give me further pause about the rigor of the MS analysis. They are using a MS instrument with high mass accuracy and statements such as "very close" and "slightly altering column performance" can easily be address with further MS analysis. I do realize that this might be asking for more work to be carried out (which may be hard to do) but qualifiers for MS results, where structural analysis and new columns shouldn't be an impedance, should be possible.

- 1) The m/z values of the molecules we detect are very close to the theoretical values and are consistent with our purified authentic standard for the substrate
- 2) Drift in the retention times of the PaLpxC substrate and product between samples is likely a result of the samples slightly altering column performance.

RESPONSE: We thank the reviewer for their thoughts on this experiment. We have repeated the analysis and obtained fragmentation data for the peaks in question such that they can be confidently assigned as the LpxC substrate and product. The new results are presented in the revised Extended Data Figure 8.

Reviewer Reports on the Third Revision:

Referee #2 (Remarks to the Author):

The authors have adequately addressed the reviewer critiques. Congratulations on an excellent study!

Referee #4 (Remarks to the Author):

I have no further concerns about the rigor of the MS data after the addition of the results presented in Extended Data 8.